# Risk Bounds and Calibration for a Smart Predict-then-Optimize Method

**Heyuan Liu**
University of California, Berkeley
Berkeley, CA 94720
`heyuan_liu@berkeley.edu`

**Paul Grigas**
University of California, Berkeley
Berkeley, CA 94720
`pgrigas@berkeley.edu`

## Abstract

The predict-then-optimize framework is fundamental in practical stochastic decision-making problems: first predict unknown parameters of an optimization model, then solve the problem using the predicted values. A natural loss function in this setting is defined by measuring the decision error induced by the predicted parameters, which was named the Smart Predict-then-Optimize (SPO) loss by Elmachtoub and Grigas [2021]. Since the SPO loss is typically nonconvex and possibly discontinuous, Elmachtoub and Grigas [2021] introduced a convex surrogate, called the SPO+ loss, that importantly accounts for the underlying structure of the optimization model. In this paper, we greatly expand upon the consistency results for the SPO+ loss provided by Elmachtoub and Grigas [2021]. We develop risk bounds and uniform calibration results for the SPO+ loss relative to the SPO loss, which provide a quantitative way to transfer the excess surrogate risk to excess true risk. By combining our risk bounds with generalization bounds, we show that the empirical minimizer of the SPO+ loss achieves low excess true risk with high probability. We first demonstrate these results in the case when the feasible region of the underlying optimization problem is a polyhedron, and then we show that the results can be strengthened substantially when the feasible region is a level set of a strongly convex function. We perform experiments to empirically demonstrate the strength of the SPO+ surrogate, as compared to standard $\ell_1$ and squared $\ell_2$ prediction error losses, on portfolio allocation and cost-sensitive multi-class classification problems.

## 1 Introduction

The *predict-then-optimize* framework, where one predicts the unknown parameters of an optimization model and then plugs in the predictions before solving, is prevalent in applications of machine learning. Some typical examples include predicting future asset returns in portfolio allocation problems and predicting the travel time on each edge of a network in navigation problems. In most cases, there are many contextual features available, such as time of day, weather information, financial and business news headlines, and many others, that can be leveraged to predict the unknown parameters and reduce uncertainty in the decision making problem. Ultimately, the goal is to produce a high quality prediction model that leads to a good decisions when implemented, such as a position that leads to a large return or a route that induces a small realized travel time. There has been a fair amount of recent work examining this paradigm and other closely related problems in data-driven decision making, such as the works of Bertsimas and Kallus [2020], Donti et al. [2017], Elmachtoub and Grigas [2021], Kao et al. [2009], Estes and Richard [2019], Ho and Hanasusanto [2019], Notz and Pibernik [2019], Kotary et al. [2021], the references therein, and others.

In this work, we focus on the particular and important case where the optimization problem of interest has a linear objective with a known convex feasible region and where the contextual features are related to the coefficients of the linear objective. This case includes the aforementioned shortest path and portfolio allocation problems. In this context, Elmachtoub and Grigas [2021] developed the Smart Predict-then-Optimize (SPO) loss function, which directly measures the regret of a prediction against the best decision in hindsight (rather than just prediction error, such as squared error). After the introduction of the SPO loss, recent work has studied its statistical properties, including generalization bounds of the SPO loss function in El Balghiti et al. [2019] and generalization and regret convergence rates in Hu et al. [2020]. Moreover, since the SPO loss is not continuous nor convex in general [Elmachtoub and Grigas, 2021], which makes the training of a prediction model computationally intractable, Elmachtoub and Grigas [2021] introduced a novel convex surrogate loss, referred to as the SPO+ loss. Elmachtoub and Grigas [2021] highlight and prove several advantages of the SPO+ surrogate loss: *(i)* it still accounts for the downstream optimization problem when evaluating the quality of a prediction model (unlike prediction losses such as the squared $\ell_2$ loss), *(ii)* it has a desirable Fisher consistency property with respect to the SPO loss under mild conditions, and *(iii)* it often performs better than commonly considered prediction losses in experimental results. Unfortunately, although a desirable property of any surrogate loss in this context, Fisher consistency is not directly applicable when one only has available a finite dataset, which is always the case in practice, because it relies on full knowledge of the underlying distribution. Motivated thusly, it is desirable to develop *risk bounds* that allow one to translate an approximate guarantee on the risk of a surrogate loss function to a corresponding guarantee on the SPO risk. That is, risk bounds (and the related notion of calibration functions) answer the question: to what tolerance $\delta$ should the surrogate excess risk be reduced to in order to ensure that the excess SPO risk is at most $\epsilon$? Note that, with enough data, it is possible in practice to ensure a (high probability) bound on the excess surrogate risk through generalization and optimization guarantees.

The main goal of this work is to provide risk bounds for the SPO+ surrogate loss function. Our results, to the best of our knowledge, are the first risk bounds of the SPO+ loss, besides the analysis of the 1-dimensional scenario in Ho-Nguyen and Kılınç-Karzan [2020]. Our results consider two cases for the structure of the feasible region of the optimization problem: *(i)* the case of a bounded polyhedron, and *(ii)* the case of a level set of a smooth and strongly convex function. In the polyhedral case, we prove that the risk bound of the SPO+ surrogate is $O(\epsilon^2)$, where $\epsilon$ is the desired accuracy for the excess SPO risk. Our results hold under mild distributional assumptions that extend those considered in Elmachtoub and Grigas [2021]. In the strongly convex level set case, we improve the risk bound of the SPO+ surrogate to $O(\epsilon)$ by utilizing novel properties of such sets that we develop, namely stronger optimality guarantees and continuity properties. As a consequence of our analysis, we can leverage generalization guarantees for the SPO+ loss to obtain the first sample complexity bounds, with respect to the SPO risk, for the SPO+ surrogate under the two cases we consider. In Section 5, we present computational results that validate our theoretical findings. In particular we present results on entropy constrained portfolio allocation problems which, to the best of our knowledge, is the first computational study of predict-then-optimize problems for a strongly convex feasible region. Our results on portfolio allocation problems demonstrate the effectiveness of the SPO+ surrogate. We also present results for cost-sensitive multi-class classification that illustrate the benefits of faster convergence of the SPO risk in the case of strongly convex sets as compared to polyhedral ones.

Starting with binary classification, risk bounds and calibration have been previously studied in other machine learning settings. Pioneer works studying the properties of convex surrogate loss functions for the 0-1 loss include Zhang et al. [2004], Bartlett et al. [2006], Massart et al. [2006] and Steinwart [2007]. Works including Zhang [2004], Tewari and Bartlett [2007] and Osokin et al. [2017] have studied the consistency and calibration properties of multi-class classification problems, which can be considered as a special case of the predict-then-optimize framework [El Balghiti et al., 2019]. Most related to the results presented herein is the work of Ho-Nguyen and Kılınç-Karzan [2020], who study the uniform calibration properties of the squared $\ell_2$ loss, and the related work of Hu et al. [2020], who also develop fast sample complexity results for the SPO loss when using a squared $\ell_2$ surrogate.

**Notation.** Let $\odot$ represent element-wise multiplication between two vectors. For any positive integer $m$, let $[m]$ denote the set $\{1, \ldots, m\}$. Let $I_p$ denote the $p$ by $p$ identity matrix for any positive integer $p$. For $\bar{c} \in \mathbb{R}^d$ and a positive semi-definite matrix $\Sigma \in \mathbb{R}^{d \times d}$, let $\mathcal{N}(\bar{c}, \Sigma)$ denote the normal distribution $\mathbb{P}(c) = \frac{e^{-\frac{1}{2}(c-\bar{c})^T \Sigma^{-1}(c-\bar{c})}}{\sqrt{(2\pi)^d \det(\Sigma)}}$. We will make use of a generic given norm $\| \cdot \|$ on $w \in \mathbb{R}^d$,

as well as its dual norm $\| \cdot \|_*$ which is defined by $\|c\|_* = \max_{w : \|w\| \le 1} c^T w$. For a positive definite matrix $A$, we define the $A$-norm by $\|w\|_A := \sqrt{w^T A w}$. Also, we denote the diameter of the set $S \subseteq \mathbb{R}^d$ by $D_S := \sup_{w, w' \in S} \|w - w'\|_2$.

## 2 Predict-then-optimize framework and preliminaries

We now formally describe the predict-then-optimize framework, which is widely prevalent in stochastic decision making problems. We assume that the problem of interest has a linear objective, but the cost vector of the objective, $c \in \mathcal{C} \subseteq \mathbb{R}^d$, is not observed when the decision is made. Instead, we observe a feature vector $x \in \mathcal{X} \subseteq \mathbb{R}^p$, which provides contextual information associated with $c$. Let $\mathbb{P}$ denote the underlying joint distribution of the pair $(x, c)$. Let $w$ denote the decision variable and assume that we have full knowledge of the feasible region $S \subseteq \mathbb{R}^d$, which is assumed to be non-empty, compact, and convex. When a feature vector $x$ is provided, the goal of the decision maker is to solve the *contextual stochastic optimization problem:*

$$\min_{w \in S} \mathbb{E}_{c \sim \mathbb{P}(\cdot | x)}[c^T w] = \min_{w \in S} \mathbb{E}_{c \sim \mathbb{P}(\cdot | x)}[c]^T w. \tag{1}$$

As demonstrated by (1), for linear optimization problems the predict-then-optimize framework relies on a prediction of the conditional expectation of the cost vector, namely $\mathbb{E}[c|x] = \mathbb{E}_{c \sim \mathbb{P}(\cdot|x)}[c]$. Let $\hat{c}$ denote a prediction of the conditional expectation, then the next step in the predict-then-optimize setting is to solve the deterministic optimization problem with the cost vector $\hat{c}$, namely

$$P(\hat{c}) : \quad \min_{w \in S} \hat{c}^T w. \tag{2}$$

Depending on the structure of the feasible region $S$, the optimization problem $P(\cdot)$ can represent linear programming, conic programming, and even (mixed) integer programming, for example. In any case, we assume that we can solve $P(\cdot)$ to any desired accuracy via either a closed-form solution or a solver. Let $w^*(\cdot) : \mathbb{R}^d \to S$ denote a particular optimization oracle for problem (2), whereby $w^*(\hat{c})$ is an optimal solution of $P(\hat{c})$. (We assume that the oracle is deterministic and ties are broken in an arbitrary pre-specified manner.)

In order to obtain a model for predicting cost vectors, namely a cost vector predictor function $g : \mathcal{X} \to \mathbb{R}^d$, we may leverage machine learning methods to learn the underlying distribution $\mathbb{P}$ from observed data $\{(x_1, c_1), \ldots, (x_n, c_n)\}$, which are assumed to be independent samples from $\mathbb{P}$. Most importantly, following (1), we would like to learn the conditional expectation and thus $g(x)$ can be thought of as an estimate of $\mathbb{E}[c|x]$. We follow a standard recipe for learning a predictor function $g$ where we specify a loss function to measure the quality of predictions relative to the observed realized cost vectors. In particular, for a loss function $\ell$, the value $\ell(\hat{c}, c)$ represents the loss or error incurred when the cost vector prediction is $\hat{c}$ (i.e., $\hat{c} = g(x)$) and the realized cost vector is $c$. Let $R_\ell(g; \mathbb{P}) := \mathbb{E}_{(x, c) \sim \mathbb{P}}[\ell(g(x), c)]$ denote the the risk of given loss function $\ell$ and let $R_\ell^*(\mathbb{P}) = \inf_{g'} R_\ell(g'; \mathbb{P})$ denote the optimal $\ell$-risk over all measurable functions $g'$. Also, let $\hat{R}_\ell^n(g) := \frac{1}{n} \sum_{i=1}^n \ell(g(x_i), c_i)$ denote the empirical $\ell$-risk. Most commonly used loss functions are based on directly measuring the prediction error, including the (squared) $\ell_2$ and the $\ell_1$ losses. However, these losses do not take the downstream optimization task nor the structure of the feasible region $S$ into consideration. Motivated thusly, one may consider a loss function that directly measures the decision error with respect to the optimization problem (2). Elmachtoub and Grigas [2021] formalize this notion in our context of linear optimization problems with their introduction of the SPO (Smart Predict-then-Optimize) loss function, which is defined by

$$\ell_{\text{SPO}}(\hat{c}, c) := c^T w^*(\hat{c}) - c^T w^*(c),$$

where $\hat{c} \in \mathbb{R}^d$ is the predicted cost vector and $c \in \mathcal{C}$ is the realized cost vector. Due to the possible non-convexity and possible discontinuities of the SPO loss, Elmachtoub and Grigas [2021] also propose a convex surrogate loss function, the SPO+ loss, which is defined as

$$\ell_{\text{SPO+}}(\hat{c}, c) := \max_{w \in S} \{(c - 2\hat{c})^T w\} + 2\hat{c}^T w^*(c) - c^T w^*(c).$$

Importantly, the SPO+ loss still accounts for the downstream optimization problem (2) and the structure of the feasible region $S$, in contrast to losses that focus only on prediction error. As discussed by Elmachtoub and Grigas [2021], the SPO+ loss can be efficiently optimized via linear/conic

optimization reformulations and with (stochastic) gradient methods for large datasets. Elmachtoub and Grigas [2021] provide theoretical and empirical justification for the use of the SPO+ loss function, including a derivation through duality theory, promising experimental results on shortest path and portfolio optimization instances, and the following theorem which provides the Fisher consistency of the SPO+ loss.

**Theorem 2.1** (Elmachtoub and Grigas [2021], Theorem 1). *Suppose that the feasible region $S$ has a non-empty interior. For fixed $x \in \mathcal{X}$, suppose that the conditional distribution $\mathbb{P}(\cdot|x)$ is continuous on all of $\mathbb{R}^d$, is centrally symmetric around its mean $\bar{c} := \mathbb{E}_{c \sim \mathbb{P}(\cdot|x)}[c]$, and that there is a unique optimal solution of $P(\bar{c})$. Then, for all $\Delta \in \mathbb{R}^p$ it holds that*

$$\mathbb{E}_{c \sim \mathbb{P}(\cdot|x)} \left[ \ell_{\text{SPO+}}(\bar{c} + \Delta, c) - \ell_{\text{SPO+}}(\bar{c}, c) \right] = \mathbb{E}_{c \sim \mathbb{P}(\cdot|x)} \left[ (c + 2\Delta)^T (w^*(c) - w^*(c + 2\Delta)) \right] \geq 0.$$

*Moreover, if $\Delta \neq 0$, then $\mathbb{E}_{c \sim \mathbb{P}(\cdot|x)} \left[ \ell_{\text{SPO+}}(\bar{c} + \Delta) - \ell_{\text{SPO+}}(\bar{c}) \right] > 0$.*

Notice that Theorem 2.1 holds for *arbitrary* $x \in \mathcal{X}$, i.e., it employs a nonparametric analysis as is standard in consistency and calibration results, whereby there are no constraints on the predicted cost vector associated with $x$. Under the conditions of Theorem 2.1, given any $x \in \mathcal{X}$, we know that the conditional expectation $\bar{c} = \mathbb{E}_{c \sim \mathbb{P}(\cdot|x)}[c]$ is the unique minimizer of the SPO+ risk. Furthermore, since $\bar{c}$ is also a minimizer of the SPO risk, it holds that the SPO+ loss function is Fisher consistent with respect to the SPO loss function, i.e., minimizing the SPO+ risk also minimizes the SPO risk. However, in practice, due to the fact that we have available only a finite dataset and not complete knowledge of the distribution $\mathbb{P}$, we cannot directly minimize the true SPO+ risk. Instead, by employing the use of optimization and generalization guarantees, we are able to approximately minimize the SPO+ risk. A natural question is then: does a low excess SPO+ risk guarantee a low excess SPO risk? More formally, we are primarily interested in the following questions: *(i)* for any $\epsilon > 0$, does there exist $\delta(\epsilon) > 0$ such that $R_{\text{SPO+}}(g; \mathbb{P}) - R_{\text{SPO+}}^*(\mathbb{P}) < \delta(\epsilon)$ implies that $R_{\text{SPO}}(g; \mathbb{P}) - R_{\text{SPO}}^*(\mathbb{P}) < \epsilon$?, and *(ii)* what is the largest such value of $\delta(\epsilon)$ that guarantees the above?

**Excess risk bounds via calibration.** The notions of calibration and calibration functions provide a useful set of tools to answer the previous questions. We now review basic concepts concerning calibration when using a generic surrogate loss function $\ell$, although we are primarily interested in the aforementioned SPO+ surrogate. An excess risk bound allows one to transfer the conditional excess $\ell$-risk, $\mathbb{E}[\ell(\hat{c}, c)|x] - \inf_{c'} \mathbb{E}[\ell(c', c)|x]$, to the conditional excess $\ell_{\text{SPO}}$-risk, $\mathbb{E}[\ell_{\text{SPO}}(\hat{c}, c)|x] - \inf_{c'} \mathbb{E}[\ell_{\text{SPO}}(c', c)|x]$. Calibration, which we now briefly review, is a central tool in developing excess risk bounds. We adopt the definition of calibration presented by Steinwart [2007] and Ho-Nguyen and Kılınç-Karzan [2020], which is reviewed in Definition 2.1 below.

**Definition 2.1.** For a given surrogate loss function $\ell$, we say $\ell$ is $\ell_{\text{SPO}}$-calibrated with respect to $\mathbb{P}$ if there exists a function $\delta_\ell(\cdot) : \mathbb{R}_+ \to \mathbb{R}_+$ such that for all $x \in \mathcal{X}$, $\hat{c} \in \mathcal{C}$, and $\epsilon > 0$, it holds that

$$\mathbb{E}[\ell(\hat{c}, c)|x] - \inf_{c'} \mathbb{E}[\ell(c', c)|x] < \delta_\ell(\epsilon) \Rightarrow \mathbb{E}[\ell_{\text{SPO}}(\hat{c}, c)|x] - \inf_{c'} \mathbb{E}[\ell_{\text{SPO}}(c', c)|x] < \epsilon. \quad (3)$$

Additionally, if (3) holds for all $\mathbb{P} \in \mathcal{P}$, where $\mathcal{P}$ is a class of distributions on $\mathcal{X} \times \mathcal{C}$, then we say that $\ell$ is uniformly calibrated with respect to the class of distributions $\mathcal{P}$.

A direct approach to finding a feasible $\delta_\ell(\cdot)$ function and checking for uniform calibration is by computing the infimum of the excess surrogate loss subject to a constraint that the excess SPO loss is at least $\epsilon$. This idea leads to the definition of the *calibration function*, which we review in Definition 2.2 below.

**Definition 2.2.** For a given surrogate loss function $\ell$ and true cost vector distribution $\mathbb{P}_c$, the conditional calibration function $\hat{\delta}_\ell(\cdot; \mathbb{P}_c)$ is defined, for $\epsilon > 0$, by

$$\hat{\delta}_\ell(\epsilon; \mathbb{P}_c) := \inf_{\hat{c} \in \mathbb{R}^d} \left\{ \mathbb{E}[\ell(\hat{c}, c)] - \inf_{c'} \mathbb{E}[\ell(c', c)] \ : \ \mathbb{E}[\ell_{\text{SPO}}(\hat{c}, c)] - \inf_{c'} \mathbb{E}[\ell_{\text{SPO}}(c', c)] \geq \epsilon \right\}.$$

Moreover, given a class of joint distributions $\mathcal{P}$, with a slight abuse of notation, the calibration function $\hat{\delta}_\ell(\cdot; \mathcal{P})$ is defined, for $\epsilon > 0$, by

$$\hat{\delta}_\ell(\epsilon; \mathcal{P}) := \inf_{x \in \mathcal{X}, \mathbb{P} \in \mathcal{P}} \hat{\delta}_\ell(\epsilon; \mathbb{P}(\cdot|x)).$$

If the calibration function $\hat{\delta}_\ell(\cdot; \mathcal{P})$ satisfies $\hat{\delta}_\ell(\epsilon; \mathcal{P}) > 0$ for all $\epsilon > 0$, then the loss function $\ell$ is uniformly $\ell_{\text{SPO}}$-calibrated with respect to the class of distributions $\mathcal{P}$. To obtain an excess risk bound, we let $\delta_\ell^{**}$ denote the biconjugate, the largest convex lower semi-continuous envelope, of $\delta_\ell$. Jensen's inequality then readily yields $\delta_\ell^{**}(R_{\text{SPO}}(g, \mathbb{P}) - R_{\text{SPO}}^*(\mathbb{P})) \leq R_\ell(g, \mathbb{P}) - R_\ell(\mathbb{P})$, which implies that the excess surrogate risk $R_\ell(g, \mathbb{P}) - R_\ell(\mathbb{P})$ of a predictor $g$ can be translated into an upper bound of the excess SPO risk $R_{\text{SPO}}(g, \mathbb{P}) - R_{\text{SPO}}^*(\mathbb{P})$. For example, the uniform calibration of the least squares (squared $\ell_2$) loss, namely $\ell_{\text{LS}}(\hat{c}, c) = \|\hat{c} - c\|_2^2$, was examined by Ho-Nguyen and Kılınç-Karzan [2020]. They proved that the calibration function is $\delta_{\ell_{\text{LS}}}(\epsilon) = \epsilon^2/D_S^2$, which implies an upper bound of the excess SPO risk by $R_{\text{SPO}}(g, \mathbb{P}) - R_{\text{SPO}}^*(\mathbb{P}) \leq D_S(R_{\text{LS}}(g, \mathbb{P}) - R_{\text{LS}}^*(\mathbb{P}))^{1/2}$. In this paper, we derive the calibration function of the SPO+ loss and thus reveal the quantitative relationship between the excess SPO risk and the excess surrogate SPO+ risk under different circumstances.

## 3    Risk bounds and calibration for polyhedral sets

In this section, we consider the case when the feasible region $S$ is a bounded polyhedron and derive the calibration function of the SPO+ loss function. As is shown in Theorem 2.1, the SPO+ loss is Fisher consistent when the conditional distribution $\mathbb{P}(\cdot|x)$ is continuous on all of $\mathbb{R}^d$ and is centrally symmetric about its mean $\bar{c}$. More formally, the joint distribution $\mathbb{P}$ lies in the distribution class $\mathcal{P}_{\text{cont, symm}} := \{\mathbb{P} : \mathbb{P}(\cdot|x) \text{ is continuous on all of } \mathbb{R}^d \text{ and is centrally symmetric about its mean, for all } x \in \mathcal{X}\}$. In Example 1, we later show that this distribution class is not restrictive enough to obtain a meaningful calibration function. Instead, we consider a more specific distribution class consisting of distributions whose density functions can be lower bounded by a normal distribution. More formally, for given parameters $M \geq 1$ and $\alpha, \beta > 0$, define $\mathcal{P}_{M,\alpha,\beta} := \{\mathbb{P} \in \mathcal{P}_{\text{cont, symm}} : \text{ for all } x \in \mathcal{X} \text{ with } \bar{c} = \mathbb{E}[c|x], \text{ there exists } \sigma \in [0, M] \text{ satisfying } \|\bar{c}\|_2 \leq \beta\sigma \text{ and } \mathbb{P}(c|x) \geq \alpha \cdot \mathcal{N}(\bar{c}, \sigma^2 I) \text{ for all } c \in \mathbb{R}^d\}$. Intuitively, the assumptions on the distribution class $\mathcal{P}_{M,\alpha,\beta}$ ensure that we avoid a situation where the density of the cost vector concentrates around some "badly behaved points." This intuition is further highlighted in Example 1. Theorem 3.1 is our main result in the polyhedral case and demonstrates that the previously defined distribution class is a sufficient class to obtain a positive calibration function. Recall that $D_S$ denotes the diameter of $S$ and define a "width constant" associated with $S$ by $d_S := \min_{v \in \mathbb{R}^d : \|v\|_2 = 1} \{\max_{w \in S} v^T w - \min_{w \in S} v^T w\}$. Notice that $d_S > 0$ whenever $S$ has a non-empty interior.

**Theorem 3.1.** *Suppose that the feasible region $S$ is a bounded polyhedron and define $\Xi_S := (1 + \frac{2\sqrt{3}D_S}{d_S})^{1-d}$. Then the calibration function of the SPO+ loss satisfies*

$$\hat{\delta}_{\ell_{\text{SPO+}}}(\epsilon; \mathcal{P}_{M,\alpha,\beta}) \geq \frac{\alpha \Xi_S}{4\sqrt{2\pi}e^{\frac{3(1+\beta^2)}{2}}} \cdot \min\left\{\frac{\epsilon^2}{D_S M}, \epsilon\right\} \quad \text{for all } \epsilon > 0. \tag{4}$$

Theorem 3.1 yields an $O(\epsilon^2)$ uniform calibration result for the distribution class $\mathcal{P}_{M,\alpha,\beta}$. The dependence on the constants is also natural as it matches the upper bound given by the cases with a $\ell_1$-like unit ball feasible region $S$ and standard multivariate normal distribution as the conditional probability $\mathbb{P}(\cdot|x)$. Please refer to Example 2 in the Appendix for a detailed discussion. Let us now provide some more intuition on the parameters involved in the definition of the distribution class $\mathcal{P}_{M,\alpha,\beta}$ and their roles in Theorem 3.1. In the definition of $\mathcal{P}_{M,\alpha,\beta}$, $\alpha$ is a lower bound on the ratio of the density of the distribution of the cost vector relative to a "reference" standard normal distribution. When $\alpha$ is larger, the distribution is behaved more like a normal distribution and it leads to a better lower bound on the calibration function (4) in Theorem 3.1. The parameter $M$ is an upper bound on the standard deviation of the aforementioned reference normal distribution, and the lower bound (4) naturally becomes worse as $M$ increases. The parameter $\beta$ measures how the conditional mean deviates from zero relative to the standard deviation of the reference normal distribution. If this distance is larger then the predictions are larger on average and (4) becomes worse. The width constant $d_S$ measures the near-degeneracy of the polyhedron ($d_S = 0$ is degenerate) and the bound becomes meaningless as $d_S \to 0$. When the feasible region $S$ is near-degenerate, i.e., the ratio $\frac{d_S}{D_S}$ is close to zero, we tend to have a weaker lower bound on the calibration function, which is also natural.

We now state a remark concerning an extension of Theorem 3.1 and we describe the example that demonstrates that it is not sufficient to consider the more general distribution class $\mathcal{P}_{\text{cont, symm}}$.

*Remark.* In Theorem 3.1, we assume that the conditional distribution $\mathbb{P}(\cdot|x)$ is lower bounded by a normal density on the entire space $\mathbb{R}^d$. We can extend the result of Theorem 3.1 to the case when

$\mathbb{P}(\cdot|x)$ is lower bounded by a normal density on a bounded set but the constant is more involved. For details, please refer to Theorem B.1 in the Appendix.

*Example* 1. Let the feasible region be the $\ell_1$ ball $S = \{w \in \mathbb{R}^2 : \|w\|_1 \leq 1\}$ and consider the distribution class $\mathcal{P}_{\text{cont, symm}}$. For a fixed scalar $\epsilon > 0$, let $c_1 = (9\epsilon, 0)^T$ and $c_2 = (-7\epsilon, 0)^T$. Let the conditional distribution $\mathbb{P}_\sigma(c|x)$ be a mixture of Gaussians defined by $\mathbb{P}_\sigma(c|x) := \frac{1}{2}(\mathcal{N}(c_1, \sigma^2 I) + \mathcal{N}(c_2, \sigma^2 I))$ for any $\sigma > 0$, and we have $\mathbb{P}_\sigma(c|x) \in \mathcal{P}_{\text{cont, symm}}$. Let the predicted cost vector be $\hat{c} = (0, \epsilon)^T$, then the excess conditional SPO risk is $\mathbb{E}[\ell_{\text{SPO}}(\hat{c}, c) - \ell_{\text{SPO}}(\bar{c}, c)|x] = \epsilon$. Then it holds that the excess conditional SPO+ risk $\mathbb{E}[\ell_{\text{SPO+}}(\hat{c}, c) - \ell_{\text{SPO+}}(\bar{c}, c)|x] \to 0$ when $\sigma \to 0$, and hence we have $\hat{\delta}_\ell(\epsilon; \mathcal{P}_{\text{cont, symm}}) = 0$.

The intuition of Example 1 is that the existence of a non-zero calibration function requires the conditional distribution of $c$ given $x$ to be "uniform" on the space $\mathbb{R}^d$, but not concentrate near certain points. Example 1 highlights a situation that considers one such "badly behaved" case where a limiting distribution of a mixture of two Gaussians leads to a zero calibration function.

By combining Theorem 3.1 with a generalization bound for the SPO+ loss, we can develop a sample complexity bound with respect to the SPO loss. Corollary 3.1 below presents such a result for the SPO+ method with a polyhedral feasible region. The derivation of Corollary 3.1 relies on the notion of *multivariate Rademacher complexity* as well as the vector contraction inequality of Maurer [2016] in the $\ell_2$-norm. In particular, for a hypothesis class $\mathcal{H}$ of cost vector predictor functions (functions from $\mathcal{X}$ to $\mathbb{R}^d$), the multivariate Rademacher complexity is defined as $\mathfrak{R}^n(\mathcal{H}) = \mathbb{E}_{\boldsymbol{\sigma}, x}\left[\sup_{g \in \mathcal{H}} \frac{1}{n} \sum_{i=1}^n \boldsymbol{\sigma}_i^T g(x_i)\right]$, where $\boldsymbol{\sigma}_i \in \{-1, +1\}^d$ are Rademacher random vectors for $i = 1, \ldots, n$. Please refer to Appendix A for a detailed discussion of multivariate Rademacher complexity and the derivation of Corollary 3.1.

**Corollary 3.1.** *Suppose that the feasible region $S$ is a bounded polyhedron, the optimal predictor $g^*(x) = \mathbb{E}[c|x]$ is in the hypothesis class $\mathcal{H}$, and there exists a constant $C'$ such that $\mathfrak{R}^n(\mathcal{H}) \leq \frac{C'}{\sqrt{n}}$.*

*Let $\hat{g}_{\text{SPO+}}^n$ denote the predictor which minimizes the empirical SPO+ risk $\hat{R}_{\text{SPO+}}^n(\cdot)$ over $\mathcal{H}$. Then there exists a constant $C$ such that for any $\mathbb{P} \in \mathcal{P}_{M,\alpha,\beta}$ and $\delta \in (0, \frac{1}{2})$, with probability at least $1 - \delta$, it holds that*

$$R_{\text{SPO}}(\hat{g}_{\text{SPO+}}^n; \mathbb{P}) - R_{\text{SPO}}^*(\mathbb{P}) \leq \frac{C\sqrt{\log(1/\delta)}}{n^{1/4}}.$$

Notice that the rate in Corollary 3.1 is $O(1/n^{1/4})$. However, the bound is with respect to the SPO loss which is generally non-convex and is the first such bound for the SPO+ surrogate. Hu et al. [2020] present a similar result for the squared $\ell_2$ surrogate with a rate of $O(1/\sqrt{n})$, and an interesting open question concerns whether the rate can also be improved for the SPO+ surrogate.

## 4 Risk bounds and calibration for strongly convex level sets

In this section, we develop improved risk bounds for the SPO+ loss function under the assumption that the feasible region is the level set of a strongly convex and smooth function, formalized in Assumption 4.1 below.

**Assumption 4.1.** *For a given norm $\|\cdot\|$, let $f : \mathbb{R}^d \to \mathbb{R}$ be a $\mu$-strongly convex and $L$-smooth function for some $L \geq \mu > 0$. Assume that the feasible region $S$ is defined by $S = \{w \in \mathbb{R}^d : f(w) \leq r\}$ for some constant $r > f_{\min} := \min_w f(w)$.*

The results in this section actually hold in more general situations than Assumption 4.1. In Appendix C, we extend the results of this section to allow the domain of the strongly convex and smooth function in Assumption 4.1 to be any set defined by linear equalities and convex inequalities (Assumption C.1). Herein, we consider the simplified case where the domain is $\mathbb{R}^d$ for ease of exposition and conciseness. The results of this section and the extension developed in Appendix C allow for a broad choice of feasible regions, for instance, any bounded $\ell_q$ ball for any $q \in (1, 2]$ and the probability simplex with entropy constraint. The latter example, which can also be thought of as portfolio allocation with an entropy constraint, is considered in the experiments in Section 5.

As in the polyhedral case, the distribution class $\mathcal{P}_{\text{cont, symm}}$ is not restrictive enough to derive a meaningful lower bound on the calibration function of the SPO+ loss. We instead consider two related classes of rotationally symmetric distributions with bounded conditional coefficient of variation.

These distribution classes are formally defined in Definition 4.1 below, and include the multi-variate Gaussian, Laplace, and Cauchy distributions as special cases.

**Definition 4.1.** Let $A$ be a given positive definite matrix. We define $\mathcal{P}_{\text{rot symm},A}$ as the class of distributions with conditional rotational symmetry in the norm $\|\cdot\|_{A^{-1}}$, namely

$$\mathcal{P}_{\text{rot symm},A} := \{\mathbb{P} : \forall x \in \mathcal{X}, \exists q(\cdot) : [0,\infty] \to [0,\infty] \text{ such that } \mathbb{P}(c|x) = q(\|c - \bar{c}\|_{A^{-1}})\}.$$

Let $\bar{c}$ denote the conditional expectation $\bar{c} = \mathbb{E}[c|x]$. For constants $\alpha \in (0,1]$ and $\beta > 0$, define

$$\mathcal{P}_{\beta,A} := \left\{\mathbb{P} \in \mathcal{P}_{\text{rot symm},A} : \mathbb{E}_{c|x}[\|c - \bar{c}\|_{A^{-1}}^2] \le \beta^2 \cdot \|\bar{c}\|_{A^{-1}}^2, \forall x \in \mathcal{X}\right\},$$

and

$$\mathcal{P}_{\alpha,\beta,A} := \left\{\mathbb{P} \in \mathcal{P}_{\text{rot symm},A} : \mathbb{P}_{c|x}(\|c - \bar{c}\|_{A^{-1}} \le \beta \cdot \|\bar{c}\|_{A^{-1}}) \ge \alpha, \forall x \in \mathcal{X}\right\}.$$

Under the above assumptions, Theorem 4.1 demonstrates that the calibration function of the SPO+ loss is $O(\epsilon)$, significantly strengthening our result in the polyhedral case. Theorem C.1 in Appendix C extends the result of Theorem 4.1 to the aforementioned case where the domain of $f(\cdot)$ may be a subset of $\mathbb{R}^d$, which includes the entropy constrained portfolio allocation problem for example.

**Theorem 4.1.** *Suppose that Assumption 4.1 holds with respect to the norm $\|\cdot\|_A$ for some positive definite matrix $A$. Then, for any $\epsilon > 0$, it holds that $\hat{\delta}_{\ell_{\text{SPO+}}}(\epsilon; \mathcal{P}_{\beta,A}) \ge \frac{\mu^{9/2}}{4(1+\beta^2)L^{9/2}} \cdot \epsilon$ and $\hat{\delta}_{\ell_{\text{SPO+}}}(\epsilon; \mathcal{P}_{\alpha,\beta,A}) \ge \frac{\alpha \mu^{9/2}}{4(1+\beta^2)L^{9/2}} \cdot \epsilon.$*

Let us now provide some more intuition on the parameters involved in the definitions of the distribution classes $\mathcal{P}_{\beta,A}$ and $\mathcal{P}_{\alpha,\beta,A}$ and their roles in Theorem 4.1. In both cases, $\beta$ controls the concentration of the distribution of cost vector around the conditional mean. The more concentrated the distribution is, the better the bounds in Theorem 4.1 are. In the case of $\mathcal{P}_{\alpha,\beta,A}$, $\alpha$ relates to the probability that the cost vector is "relatively close" to the conditional mean. When $\alpha$ is larger, the cost vector is more likely to be close to the conditional mean and the bound will be better.

Our analysis for the calibration function (the proof of Theorem 4.1) relies on the following two lemmas, which utilize the special structure of strongly convex level sets to strengthen the first-order optimality guarantees and derive a "Lipschitz-like" continuity property of the optimization oracle $w^*(\cdot)$. The first such lemma strengthens the optimality guarantees of (2) and provides both upper and lower bounds of the SPO loss.

**Lemma 4.1.** *Suppose that Assumption 4.1 holds with respect to a generic norm $\|\cdot\|$. Then, for any $c_1, c_2 \in \mathbb{R}^d$, it holds that*

$$c_1^T(w - w^*(c_1)) \ge \frac{\mu}{2\sqrt{2L(r - f_{\min})}}\|c_1\|_* \|w - w^*(c_1)\|^2, \quad \forall w \in S,$$

*and*

$$c_1^T(w^*(c_2) - w^*(c_1)) \le \frac{L}{2\sqrt{2\mu(r - f_{\min})}}\|c_1\|_* \|w^*(c_1) - w^*(c_2)\|^2.$$

The following lemma builds on Lemma 4.1 to develop upper and lower bounds on the difference between two optimal decisions based on the difference between the two normalized cost vectors.

**Lemma 4.2.** *Suppose that Assumption 4.1 holds with respect to a generic norm $\|\cdot\|$. Let $c_1, c_2 \in \mathbb{R}^d$ be such that $c_1, c_2 \neq 0$, then it holds that*

$$\|w^*(c_1) - w^*(c_2)\| \ge \frac{\sqrt{2\mu(r - f_{\min})}}{L} \cdot \left\|\frac{c_1}{\|c_1\|_*} - \frac{c_2}{\|c_2\|_*}\right\|_*,$$

*and*

$$\|w^*(c_1) - w^*(c_2)\| \le \frac{\sqrt{2L(r - f_{\min})}}{\mu} \cdot \left\|\frac{c_1}{\|c_1\|_*} - \frac{c_2}{\|c_2\|_*}\right\|_*.$$

Note that the lower bound of $c_1^T(w - w^*(c_1))$ in Lemma 4.1 and the upper bound of $\|w^*(c_1) - w^*(c_2)\|$ in Lemma 4.2 match bounds developed by El Balghiti et al. [2019]. Indeed, although El Balghiti et al. [2019] study the more general case of strongly convex sets, the constants are the same since Theorem 12 of Journée et al. [2010] demonstrates that our set $S$ is a $\frac{\mu}{\sqrt{2L(r-f_{\min})}}$-strongly convex

set. However, the upper bounds in Lemmas 4.1 and 4.2 appear to be novel and rely on the special properties of strongly convex *level* sets. It is important to emphasize that we generally do not expect all of the bounds in Lemmas 4.1 and 4.2 to holds for polyhedral sets. Indeed, for a polyhedron the optimization oracle $w^*(\cdot)$ is generally discontinuous at cost vectors that have multiple optimal solutions. The properties in Lemmas 4.1 and 4.2 drive the proof of Theorem 4.1 and hence lead to the improvement from $O(\epsilon^2)$ in the polyhedral case to $O(\epsilon)$ in the strongly convex level set case.

By following similar arguments as in the derivation of Corlloary 3.1, Corollary 4.1 presents the sample complexity of the SPO+ method when the feasible region is a strongly convex level set.

**Corollary 4.1.** *Suppose that Assumption 4.1 holds with respect to the norm $\|\cdot\|_A$ for some positive definite matrix $A$. Suppose further that the optimal predictor $g^*(x) = \mathbb{E}[c|x]$ is in the hypothesis class $\mathcal{H}$, and there exists a constant $C'$ such that $\mathfrak{R}^n(\mathcal{H}) \leq \frac{C'}{\sqrt{n}}$. Let $\hat{g}^n_{\mathrm{SPO+}}$ denote the predictor which minimizes the empirical SPO+ risk $\hat{R}^n_{\mathrm{SPO+}}(\cdot)$ over $\mathcal{H}$. Then there exists a constant $C$ such that for any $\mathbb{P} \in \mathcal{P}_{\alpha,\beta} \cup \mathcal{P}_\beta$ and $\delta \in (0, \frac{1}{2})$, with probability at least $1 - \delta$, it holds that*

$$R_{\mathrm{SPO}}(\hat{g}^n_{\mathrm{SPO+}}; \mathbb{P}) - R^*_{\mathrm{SPO}}(\mathbb{P}) \leq \frac{C\sqrt{\log(1/\delta)}}{n^{1/2}}.$$

Notice that Corollary 4.1 improves the rate of convergence to $O(1/\sqrt{n})$ as compared to the $O(1/n^{1/4})$ rate of Corollary 3.1. This matches the rate for the squared $\ell_2$ surrogate developed by Hu et al. [2020] (though their result is in the polyhedral case).

# 5 Computational experiments

In this section, we present computational results of synthetic dataset experiments wherein we empirically examine the performance of the SPO+ loss function for training prediction models, using portfolio allocation and cost-sensitive multi-class classification problems as our problem classes. We focus on two classes of prediction models: *(i)* linear models, and *(ii)* two-layer neural networks with 256 neurons in the hidden layer. We compare the performance of the empirical minimizer of the following four different loss function: *(i)* the previously described SPO loss function (when applicable), *(ii)* the previously described SPO+ loss function, *(iii)* the least squares (squared $\ell_2$) loss function $\ell(\hat{c}, c) = \|\hat{c} - c\|_2^2$, and *(iv)* the absolute ($\ell_1$) loss function $\ell(\hat{c}, c) = \|\hat{c} - c\|_1$. For all loss functions, we use the Adam method of Kingma and Ba [2015] to train the parameters of the prediction models. Note that the loss functions *(iii)* and *(iv)* do not utilize the structure of the feasible region $S$ and can be viewed as purely learning the relationship between cost and feature vectors.

**Entropy constrained portfolio allocation.** First, we consider the portfolio allocation [Markowitz, 1952] problem with entropy constraint, where the goal is to pick an allocation of assets in order to maximize the expected return while enforcing a certain level of diversity through the use of an entropy constraint (see, e.g., Bera and Park [2008]). This application is an instance of our more general theory for strongly convex level sets on constrained domains, developed in Appendix C. Alternative formulations of portfolio allocation, including when $S$ is a polyhedron or a polyhedron intersected with an ellipsoid, have been empirically studied in previous works (see, for example Elmachtoub and Grigas [2021], Hu et al. [2020]). The objective is to minimize $c^T w$ where $c$ is the negative of the expected returns of $d$ different assets, and the feasible region is $S = \{w \in \mathbb{R}^d : w \geq 0, \sum_{i=1}^d w_i = 1, \sum_{i=1}^d w_i \log w_i \leq r\}$ where $r$ is a user-specified threshold of the entropy of portfolio $w$. Note that, due to the differentiability properties of of the optimization oracle $w^*(\cdot)$ in this case (see Lemma D.1 in the Appendix), it is possible to (at least locally) optimize the SPO loss using a gradient method even though SPO loss is not convex.

In our simulations, the relationship between the true cost vector $c$ and its auxiliary feature vector $x$ is given by $c = \phi^{\deg}(Bx) \odot \epsilon$, where $\phi^{\deg}$ is a polynomial kernel mapping of degree deg, $B$ is a fixed weight matrix, and $\epsilon$ is a multiplicative noise term. The features are generated from a standard multivariate normal distribution, we consider $d = 50$ assets, and further details of the synthetic data generation process are provided in Appendix D. To account for the differing distributions of the magnitude of the cost vectors, in order to evaluate the performance of each method we compute a "normalized" SPO loss on the test set. Specifically, let $\hat{g}$ denote a trained prediction model and let $\{\tilde{x}_i, \tilde{c}_i\}_{i=1}^m$ denote the test set, then the normalized SPO loss is defined as $\frac{\sum_{i=1}^m \ell_{\mathrm{SPO}}(\hat{g}(\tilde{x}_i), \tilde{c}_i)}{\sum_{i=1}^m z^*(\tilde{c}_i)}$, where

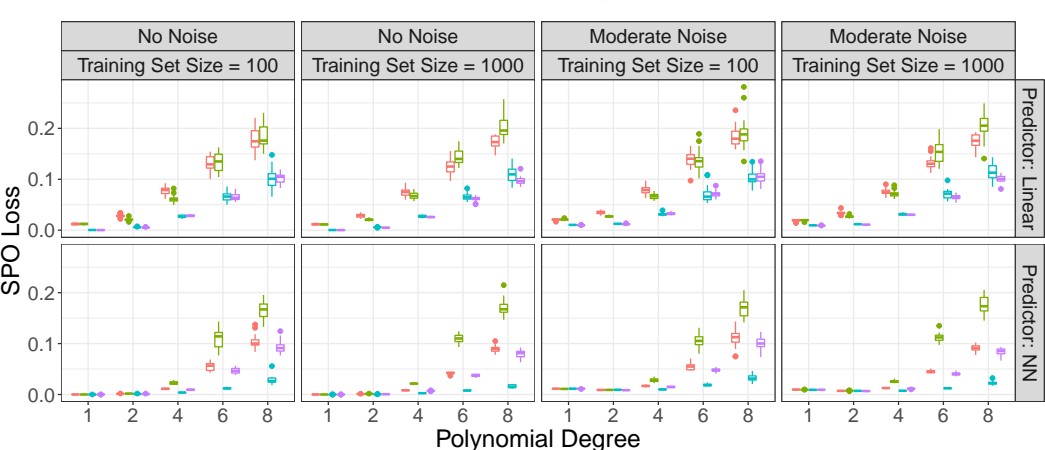

Figure 1: Normalized test set SPO loss for the SPO, SPO+, least squares, and absolute loss methods on portfolio allocation instances.

$z^*(\tilde{c}) := \min_{w \in S} \tilde{c}^T w$ is the optimal cost in hindsight. We set the size of the test set to $10000$. In all of our experiments, we run $50$ independent trials for each setting of parameters.

Figure 1 displays the empirical performance of each method. We observe that with a linear hypothesis class, for smaller values of the degree parameters, i.e., deg $\in \{1, 2\}$, all four methods perform comparably, while the SPO and SPO+ methods dominate in cases with larger values of the degree parameters. With a neural net hypothesis class, we observe a similar pattern but, due to the added degree of flexibility, the SPO method dominates the cases with larger values of degree and SPO+ method is the best among all surrogate loss functions. The better results of the $\ell_1$ loss as compared to the squared $\ell_2$ loss might be explained by robustness against outliers. Appendix D also contains results showing the observed convergence of the excess SPO risk, in the case of polynomial degree one, for both this experiment and the cost-sensitive multi-class classification case.

**Cost-sensitive multi-class classification.** Here we consider the cost-sensitive multi-class classification problem. Since this is a multi-class classification problem, the feasible region is simply the unit simplex $S = \{w \in \mathbb{R}^d : w \geq 0, \sum_{i=1}^d w_i = 1\}$ We consider an alternative model for generating the data, whereby the relationship between the true cost vector $c$ and its auxiliary feature vector $x$ is as follows: first we generate a score $s = \sigma(\phi^{\text{deg}}(b^T x) \odot \epsilon)$, where $\phi^{\text{deg}}$ is a degree-deg polynomial kernel, $b$ is a fixed weight vector, $\epsilon$ is a multiplicative noise term, and $\sigma(\cdot)$ is the logistic function. Then the true label is given by lab $= \lceil 10s \rceil \in \{1, \dots, 10\}$ and the cost vector $c$ is given by $c_i = |i - \text{lab}|$ for $i = 1, \dots, 10$. The features are generated from a standard multivariate normal distribution, and further details of the synthetic data generation process are provided in Appendix D. Since the scale of the cost vectors do not change as we change different parameters, we simply compare the test set SPO loss for each method. We still set the size of the test set to $10000$ and we run $50$ independent trials for each setting of parameters. In addition to the regular SPO+, least squares, and absolute losses, we consider an alternative surrogate loss constructed by considering the SPO+ loss using a log barrier (strongly convex) *approximation* to the unit simplex. That is, we consider the SPO+ surrogate that arises from the set $\tilde{S} := \{w \in \mathbb{R}^d : w \geq 0, \sum_{i=1}^d w_i = 1, -\sum_{i=1}^d \log w_i \leq r\}$ for some $r > 0$. Details about how we chose the value of $r$ are provided in Appendix D.

Herein we focus on the comparison between the standard SPO+ loss and the "SPO+ w/ Barrier" surrogate loss. (We include a more complete comparison of all the method akin to Figure 1 in Appendix D.) Figure 2 shows a detailed comparison between these alternative SPO+ surrogates as we vary the training set size. Note that the SPO loss is always measured with respect to the standard unit simplex and not the log barrier approximation. Interestingly, we observe that the "SPO+ w/ Barrier" surrogate tends to perform better than the regular SPO+ surrogate when the training set size is small, whereas the regular SPO+ surrogate gradually performs better as the training set size increases. These

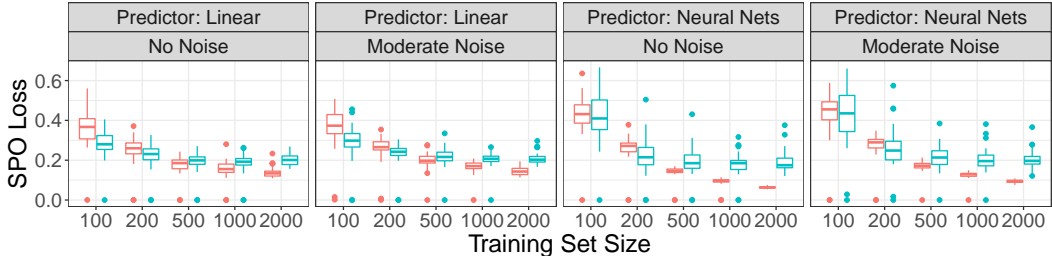

Figure 2: Test set SPO loss for the SPO+ methods with different feasible regions on the cost-sensitive multi-class classification instances.

results suggest that adding a barrier constraint to the feasible region has a type of regularization effect, which may also be explained by our theoretical results. Indeed, adding the barrier constraint makes the feasible region strongly convex, which improves the rate of convergence of the SPO risk. On the other hand, this results in an approximation to the actual feasible region of interest and, eventually for large enough training set sizes, the regularizing benefit of the barrier constraint is outweighed by the cost of this approximation.

## 6 Conclusions and future directions

Our work develops risk bounds and uniform calibration results for the SPO+ loss relative to the SPO loss, and our results provide a quantitative way to transfer the excess surrogate risk to excess true risk. We analyze the case with a polyhedral feasible region of the underlying optimization problem, and we strengthen the results when the feasible region is a level set of a strongly convex function. There are several intriguing future directions. In this work, we mainly focus on the worst case risk bounds of the SPO+ method under different types of feasible regions, and we consider the minimal possible conditions to guarantee the risk bounds. In many practical problems, it is reasonable to assume the true joint distribution of $(x, c)$ satisfies certain "low-noise" conditions (see, for example, Bartlett et al. [2006], Massart et al. [2006], Hu et al. [2020]). In such conditions, one might be able to obtain improved risk bounds and sample complexities. Also, developing tractable surrogates and a corresponding calibration theory for non-linear objectives is very worthwhile.

**Acknowledgments**

The authors are grateful to Othman El Balghiti, Adam N. Elmachtoub, and Ambuj Tewari for early discussions related to this work. PG acknowledges the support of NSF Awards CCF-1755705 and CMMI-1762744.

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
