# A Rademacher complexity and generalization bounds

Herein we briefly review *Rademacher complexity*, a widely used concept in deriving generalization bounds, and how it applies in our analysis. For any loss function $\ell(\cdot, \cdot)$ and a hypothesis class $\mathcal{H}$ of cost vector predictor functions, the Rademacher complexity is defined as

$$\mathfrak{R}_\ell^n(\mathcal{H}) := \mathbb{E}_{\sigma, \{(x_i, c_i)\}_{i=1}^n} \left[ \sup_{g \in \mathcal{H}} \frac{1}{n} \sum_{i=1}^n \sigma_i \ell(g(x_i), c_i) \right],$$

where $\sigma_i$ are independent Rademacher random variables and $(x_i, c_i)$ are independent samples from the joint distribution $\mathbb{P}$ for $i = 1, \ldots, n$. The following theorem provides a classical generalization bounds based on the Rademacher complexity.

**Theorem A.1** (Bartlett and Mendelson [2002]). *Let $\mathcal{H}$ be a hypothesis class from $\mathcal{X}$ to $\mathbb{R}^d$ and let $b = \sup_{\hat{c} \in \mathcal{H}(\mathcal{X}), c \in \mathcal{C}} \ell(\hat{c}, c)$. Then, for any $\delta > 0$, with probability at least $1 - \delta$, for all $g \in \mathcal{H}$ it holds that*

$$\left| R_\ell(g; \mathbb{P}) - \hat{R}_\ell^n(g) \right| \leq 2\mathfrak{R}_\ell^n(\mathcal{H}) + b \sqrt{\frac{2 \log(1/\delta)}{n}}.$$

Moreover, we define the *multivariate Rademacher complexity* [Maurer, 2016, Bertsimas and Kallus, 2020, El Balghiti et al., 2019] of $\mathcal{H}$ as

$$\mathfrak{R}^n(\mathcal{H}) = \mathbb{E}_{\boldsymbol{\sigma}, x} \left[ \sup_{g \in \mathcal{H}} \frac{1}{n} \sum_{i=1}^n \boldsymbol{\sigma}_i^T g(x_i) \right],$$

where $\boldsymbol{\sigma}_i \in \{-1, +1\}^d$ are Rademacher random vectors for $i = 1, \ldots, n$. In many cases of hypothesis classes, such as linear functions with bounded Frobenius or element-wise $\ell_1$ norm, the multivariate Rademacher complexity can be bounded as $\mathfrak{R}^n(\mathcal{H}) \leq \frac{C'}{\sqrt{n}}$ where $C'$ is a constant that usually depends on the properties of the data, the hypothesis class, and mildly on the dimensions $d$ and $p$. Detailed examples of such bounds have been provided by El Balghiti et al. [2019], Bertsimas and Kallus [2020].

When the loss function $\ell(\cdot, \cdot)$ is additionally $L$-Lipschitz continuous with respect to the 2-norm in the first argument, namely $|\ell(\hat{c}_1, c) - \ell(\hat{c}_2, c)| \leq L \|\hat{c}_1 - \hat{c}_2\|_2$ for all $\hat{c}_1, \hat{c}_2, c \in \mathbb{R}^p$, then by the vector contraction inequality of Maurer [2016] we have $\mathfrak{R}_\ell^n(\mathcal{H}) \leq \sqrt{2} L \mathfrak{R}^n(\mathcal{H})$. It is also easy to see that the the SPO+ loss function $\ell_{\text{SPO+}}(\cdot, c)$ is $2D_S$-Lipschitz continuous with respect to the 2-norm for any $c$ and therefore we can leverage the vector contraction inequality of Maurer [2016] in this case. Combined with Theorem A.1, this yields a generalization bound for the SPO+ loss which, when combined with Theorems 3.1 and 4.1 yields Corollaries 3.1 and 4.1, respectively. The full proofs of these corollaries are included below.

*Proof of Corollary 3.1 and 4.1.* Let $b = \sup_{\hat{c} \in \mathcal{H}(\mathcal{X}), c \in \mathcal{C}} \ell(\hat{c}, c) \leq 2D_S \sup_{g \in \mathcal{H}, x \in \mathcal{X}} \|g(x)\|_2$. For any $\delta > 0$, with probability at least $1 - \delta$, for all $g \in \mathcal{H}$, it holds that

$$\left| R_\ell(g; \mathbb{P}) - \hat{R}_\ell^n(g) \right| \leq 4\sqrt{2} D_S \mathfrak{R}^n(\mathcal{H}) + b \sqrt{\frac{2 \log(1/\delta)}{n}}.$$

Since $\mathfrak{R}^n(\mathcal{H}) \leq \frac{C'}{\sqrt{n}}$ and $\log(1/\delta) \geq \log(2)$, we know that there exists some universal constant $C_1$ such that

$$4\sqrt{2} D_S \mathfrak{R}^n(\mathcal{H}) + b \sqrt{\frac{2 \log(1/\delta)}{n}} \leq C_1 \sqrt{\frac{\log(1/\delta)}{n}},$$

for all $\delta \in (0, \frac{1}{2})$ and $n \geq 1$. Since $\hat{g}_{\text{SPO+}}^n$ minimizes the empirical SPO+ risk $\hat{R}_{\text{SPO+}}^n(\cdot)$, we have $\hat{R}_{\text{SPO+}}^n(\hat{g}_{\text{SPO+}}^n) \leq \hat{R}_{\text{SPO+}}^n(g_{\text{SPO+}}^*)$. and therefore, with probability at least $1 - \delta$, it holds that

$$R_{\text{SPO+}}(\hat{g}_{\text{SPO+}}^n) - R_{\text{SPO+}}^* \leq 2C_1 \sqrt{\frac{\log(1/\delta)}{n}}.$$

Recall Theorem 3.1, the biconjugate of $\min\{\frac{\epsilon^2}{D_S M}, \epsilon\}$ is $\frac{\epsilon^2}{D_S M}$ for $\epsilon \in [0, \frac{D_S M}{2}]$ and $\epsilon - \frac{D_S M}{4}$ for $\epsilon \in [\frac{D_S M}{2}, \infty]$. Then if the assumption in Corollary 3.1 holds, with probability at least $1 - \delta$, it holds that

$$R_{\text{SPO}}(\hat{g}_{\text{SPO+}}^n; \mathbb{P}) - R_{\text{SPO}}^*(\mathbb{P}) \leq \frac{C_2 \sqrt{\log(1/\delta)}}{n^{1/4}},$$

for some universal constant $C_2$. Also, since the calibration function in Theorem C.1 is linear and thus convex, then if the assumption in Corollary 3.1 holds, with probability at least $1 - \delta$, it holds that

$$R_{\text{SPO}}(\hat{g}_{\text{SPO+}}^n; \mathbb{P}) - R_{\text{SPO}}^*(\mathbb{P}) \leq \frac{C_3 \sqrt{\log(1/\delta)}}{n^{1/2}},$$

for some universal constant $C_3$.

$\square$

## B  Proofs and other technical details for Section 3

### B.1  Additional definitions and notation

Recall that $S$ is polyhedral and let $Z_S$ denote the extreme points of $S$. We assume, for simplicity, that $w^*(c) \in Z_S$ for all $c \in \mathbb{R}^d$, but our results can be extended to allow for other possibilities in the case when there are multiple optimal solutions of $P(c)$. For any $i \in \{1, \ldots, d\}$, we use $e_i \in \mathbb{R}^d$ to represent the unit vector whose $i$-th entry is 1 and others are all zero. Given a vector $c' \in \mathbb{R}^{d-1}$ and a scalar $\xi \in \mathbb{R}$, let $(c', \xi)$ denote the vector $(c'^T, \xi)^T \in \mathbb{R}^d$. For fixed $c'$ and when $\xi$ ranges from negative infinity to positive infinity, the corresponding optimal solution $w^*(c', \xi)$ will sequentially take different values in $Z_S$, and we let $\Omega(c') = (w_1(c'), \ldots, w_{k(c')}(c'))$ denote this sequence. Let $y_i(c')$ denote the last element of vector $w_i(c')$ for $i = 1, \ldots, k(c')$. Also, for $i = 1, \ldots, k(c') - 1$, we define phase transition location $\zeta_i(c') \in \mathbb{R}$ such that $(c', \zeta_i(c'))^T w_i(c') = (c', \zeta_i(c'))^T w_{i+1}(c')$, and additionally, we define $\zeta_0(c') = -\infty$ and $\zeta_{k(c')}(c') = \infty$. When there is no confusion, we will omit $c'$ and only use $k, w_i, y_i, \zeta_i$ for simplicity.

Based on the above definition, for all $\xi \in (\zeta_{i-1}(c'), \zeta_i(c'))$, it holds that $w^*(c', \xi) = w_i(c')$. Also, it holds that $y_1(c') > \cdots > y_{k(c')}(c')$.

### B.2  Detailed derivation for Example 1

Let the feasible region be the $\ell_1$ ball $S = \{w \in \mathbb{R}^2 : \|w\|_1 \leq 1\}$ and consider the distribution class $\mathcal{P}_{\text{cont, symm}}$. Let $x \in \mathcal{X}$ be fixed, $\epsilon > 0$ be a fixed scalar, $c_1 = (9\epsilon, 0)^T$ and $c_2 = (-7\epsilon, 0)^T$. Let the conditional distribution be a mixture of normals defined by $\mathbb{P}_\sigma(c|x) := \frac{1}{2}(\mathcal{N}(c_1, \sigma^2 I) + \mathcal{N}(c_2, \sigma^2 I))$ for some $\sigma > 0$. The condition mean of $c$ is then $\bar{c} = (\epsilon, 0)^T$ and the distribution $\mathbb{P}_\sigma(c|x)$ is centrally symmetric around $\bar{c}$; therefore $\mathbb{P}_\sigma \in \mathcal{P}_{\text{cont, symm}}$. Let $\hat{c} = (0, \epsilon)^T$ and $\Delta := \hat{c} - \bar{c}$, which yields that the excess conditional SPO risk is $\mathbb{E}[\ell_{\text{SPO}}(\hat{c}, c) - \ell_{\text{SPO}}(\bar{c}, c)] = \bar{c}^T(w^*(\hat{c}) - w^*(\bar{c})) = \epsilon$. Also, for all $c \in \mathcal{C}$, we may assume that $w^*(c) \in Z_S = \{\pm e_1, \pm e_2\}$ and hence $(c + 2\Delta)^T(w^*(c) - w^*(c + 2\Delta)) \leq 2\Delta^T(w^*(c) - w^*(c + 2\Delta)) \leq 4\epsilon$. Therefore, using $\mathbb{E}[\ell_{\text{SPO+}}(\bar{c} + \Delta, c) - \ell_{\text{SPO+}}(\bar{c}, c)] = \mathbb{E}[(c + 2\Delta)^T(w^*(c) - w^*(c + 2\Delta))]$, it holds that

$$\mathbb{E}[\ell_{\text{SPO+}}(\bar{c} + \Delta, c) - \ell_{\text{SPO+}}(\bar{c}, c)] \leq 4\epsilon \mathbb{P}_\sigma(w^*(c) \neq w^*(c + 2\Delta))$$
$$\leq 4\epsilon(1 - \mathbb{P}_\sigma(\{\|c - c_1\|_2 \leq \epsilon\} \cup \{\|c - c_2\|_2 \leq \epsilon\})) \to 0,$$

when $\sigma \to 0$, and hence we have $\hat{\delta}_\ell(\epsilon; \mathcal{P}_{\text{cont, symm}}) = 0$.

### B.3  Proofs and useful lemmas

Lemma B.1 provides the relationship between excess SPO risk and the optimal solution of (2) with respect to the difference $\Delta = \hat{c} - \bar{c}$ between the predicted cost vector $\hat{c}$ and the realized cost vector $\bar{c}$.

**Lemma B.1.** *Let $\hat{c}, \bar{c} \in \mathbb{R}^d$ be given and define $\Delta := \hat{c} - \bar{c}$. Let $w_+ := w^*(\Delta)$ and $w_- := w^*(-\Delta)$, and let $y_+$ and $y_-$ denote the last elements of $w_+$ and $w_-$, respectively. If $\bar{c}^T(w^*(\hat{c}) - w^*(\bar{c})) \geq \epsilon$, then it holds that $\Delta^T(w_- - w_+) \geq \epsilon$. Additionally, if $\Delta = \kappa \cdot e_d$ for some $\kappa > 0$, then it holds that $(y_- - y_+)\kappa \geq \epsilon$.*

*Proof of Lemma B.1.* First we have $\hat{c}^T(w^*(\bar{c}) - w^*(\hat{c})) \geq 0$, and therefore it holds that $\Delta^T(w^*(\bar{c} + \Delta) - w^*(\bar{c})) \geq \bar{c}^T(w^*(\bar{c} + \Delta) - w^*(\bar{c})) \geq \epsilon$. Also, since $\Delta^T(w^*(\bar{c}) - w^*(\Delta)) \geq 0$ and $\Delta^T(w^*(-\Delta) - w^*(\bar{c} + \Delta)) \geq 0$, we have $\Delta^T(w_- - w_+) \geq \Delta^T(w^*(\bar{c} + \Delta) - w^*(\bar{c})) \geq \epsilon$. Moreover, when $\Delta = \kappa \cdot e_d$ for $\kappa > 0$, we have $\Delta^T w_- = \Delta^T w_1$ and $\Delta^T w_+ = \Delta^T w_k$, and therefore, it holds that $(y_- - y_+)\kappa \geq \epsilon$. $\square$

Lemma B.2 and B.3 provide two useful inequalities.

**Lemma B.2.** *Suppose that $a_1, \ldots, a_n, b_1, \ldots, b_n \geq 0$ with $\sum_{i=1}^n a_i = \alpha$ and $\sum_{i=1}^n b_i = \beta$ for some $\alpha, \beta > 0$. Then for all $p \geq 0$, it holds that*

$$\sum_{i=1}^n b_i \left(1 + \frac{a_i^2}{b_i^2}\right)^{-p/2} \geq \frac{\beta}{(1 + \frac{\alpha}{\beta})^p}.$$

*Proof.* Let $\psi_i(a, b; p) = b_i(1 + \frac{a_i^2}{b_i^2})^{-p/2}$ and $\psi(a, b; p) = \sum_{i=1}^n \psi_i(a, b; p)$. For all $p \in \mathbb{R}$, we have

$$\frac{\mathrm{d}^2}{\mathrm{d}p^2} \log(\psi(a, b; p)) = \frac{1}{4\psi^2(a, b; p)} \left( \sum_{i=1}^n \psi_i(a, b; p) \cdot \sum_{i=1}^n \psi_i(a, b; p) \log^2\left(1 + \frac{a_i^2}{b_i^2}\right) \right.$$

$$\left. - \left( \sum_{i=1}^n \psi_i(a, b; p) \log\left(1 + \frac{a_i^2}{b_i^2}\right) \right)^2 \right) \geq 0,$$

for $p \geq 0$. Therefore, for all $p \geq 0$ it holds that

$$\log \psi(a, b; p) \geq \log \psi(a, b; 0) + p \cdot (\log \psi(a, b; 0) - \log \psi(a, b; -1)).$$

Also, we have $\psi(a, b, 0) = \beta$, and $\psi(a, b, -1) = \sum_{i=1}^n \sqrt{a_i^2 + b_i^2} \leq \sum_{i=1}^n (a_i + b_i) = \alpha + \beta$. Then, for all $p \geq 0$, it holds that $\psi(a, b; p) \geq \frac{\beta^{p+1}}{(\alpha+\beta)^p} = \frac{\beta}{(1+\frac{\alpha}{\beta})^p}$. $\square$

**Lemma B.3.** *Let $\hat{c}' \in \mathbb{R}^{d-1}$ be given with $\|\hat{c}'\|_2 = 1$, and let $\{w_i(\hat{c}')\}_{i=1}^k$, $\{y_i(\hat{c}')\}_{i=1}^k$, and $\{\zeta_i(\hat{c}')\}_{i=0}^k$ be the corresponding optimal solution sequence and phase transition location sequence as described in Section B.1. Let $y_- = y_1(\hat{c}')$ and $y_+ = y_k(\hat{c}')$. Then it holds that*

$$\sum_{i=1}^{k-1} \left(1 + 3\zeta_i^2\right)^{-\frac{d-1}{2}} (y_i - y_{i+1}) \geq \Xi_{S, \hat{c}'} \cdot (y_- - y_+),$$

*where $\Xi_{S, \hat{c}'} = (1 + \frac{2\sqrt{3}D_S}{y_- - y_+})^{1-d}$.*

*Proof.* Let $w_i'$ be the first $(d-1)$ element of $w_i$. Suppose $\zeta_{s-1} \leq 0 < \zeta_s$ for some $s \in \{1, \ldots, k\}$, then it holds that $\hat{c}'^T(w_i - w_{i+1}) = -\zeta_i(y_i - y_{i+1}) \geq 0$ for $i \in \{1, \ldots, s-1\}$ and $\hat{c}'^T(w_i - w_{i+1}) = -\zeta_i(y_i - y_{i+1}) < 0$ for $i \in \{s, \ldots, k-1\}$. Therefore, we know that

$$\sum_{i=1}^{k-1} \left|\hat{c}'^T(w_i - w_{i+1})\right| = \hat{c}'^T(w_1 + w_k - 2w_s) \leq 2D_S.$$

Also, we have $\sum_{i=1}^{k-1}(y_i - y_{i+1}) = y_- - y_+$ and $|\zeta_i| = -\frac{|\hat{c}'^T(\hat{w}_i' - \hat{w}_{i+1}')|}{y_i - y_{i+1}}$. Therefore, by the result in Lemma B.2, we have

$$\sum_{i=1}^{k-1} \left(1 + 3\zeta_i^2\right)^{-\frac{d-1}{2}} (y_i - y_{i+1}) \geq \frac{y_- - y_+}{(1 + \frac{2\sqrt{3}D_S}{y_- - y_+})^{d-1}}.$$

$\square$

Lemma B.4 provide a lower bound of the conditional SPO+ risk condition on the first $(d-1)$ element of the realized cost vector.

**Lemma B.4.** *Let $c' \in \mathbb{R}^{d-1}$ be a fixed vector and $\bar{\xi} \in \mathbb{R}$, $\sigma > 0$ be fixed scalars. Let a random variable $\xi$ satisfying $\mathbb{P}(\xi) \geq \alpha \cdot \mathcal{N}(\bar{\xi}, \sigma^2)$ for all $\xi \in [-\sqrt{2D^2 - \|c'\|^2}, \sqrt{2D^2 - \|c'\|^2}]$. Let $c = (c', \xi) \in \mathbb{R}^d$, and sequence $\{w_i(c')\}_{i=0}^k$, $\{\zeta_i(c')\}_{i=0}^k$ defined as in Section B.1. Let $y_i$ denote the last element of vector $w_i$ for $i = 1, \ldots, k$. Let $m_i = \sqrt{1 + 3\|\zeta_i(c')\|^2/\|c'\|^2}$ for $i = 1, \ldots, k$. Suppose $\Delta = \kappa \cdot e_d$ for some $\kappa > 0$, then for all $\tilde{\kappa} \in [0, \kappa]$, it holds that*

$$\mathbb{E}_\xi \left[(c + 2\Delta)^T(w^*(c) - w^*(c + 2\Delta))\right] \geq \frac{\alpha\tilde{\kappa}\kappa e^{-\frac{3(\tilde{\kappa}^2 + \bar{\xi}^2)}{2\sigma^2}}}{2} \cdot \sum_{i=1}^{k-1} \frac{e^{-\frac{3\zeta_i^2(c')}{2\sigma^2}} \mathbb{1}\{\|c'\| \leq \frac{D}{m_i}\}}{\sqrt{2\pi}\sigma}(y_i - y_{i+1}).$$

*Proof of Lemma B.4.* Let $(w_1, \ldots, w_k) = \Omega(c')$ as defined in Section B.1, and suppose $w^*(c) = w_s$ and $w^*(c + 2\Delta) = w_t$ for some $s \leq t$. By the definition of $\{\xi_i(c')\}_0^k$, we know that $\xi \in [\zeta_{s-1}(c'), \zeta_s(c')]$ and $\xi + 2\kappa \in [\zeta_{t-1}(c'), \zeta_t(c')]$. Therefore, it holds that

$$(c + 2\Delta)^T(w^*(c) - w^*(c + 2\Delta)) = (c + 2\Delta)^T(w_s - w_t) = \sum_{i=s}^{t-1}(c + 2\Delta)^T(w_i - w_{i+1})$$

$$= \sum_{i=s}^{t-1}(c + 2\Delta - (c', \zeta_i(c')))^T(w_i - w_{i+1}) = \sum_{i=s}^{t-1}(\xi + 2\kappa - \zeta_i(c')) \cdot e_d^T(w_i - w_{i+1})$$

$$= \sum_{i=1}^{k-1}\mathbb{1}\{\xi \in [\zeta_i - 2\kappa, \zeta_i]\} \cdot (\xi + 2\kappa - \zeta_i(c'))(y_i - y_{i+1}),$$

where $y_i$ denotes the last element of $w_i$ for all $i = 1, \ldots, k$. When $\xi$ follows the normal distribution $\mathcal{N}(\bar{\xi}, \sigma^2)$, it holds that

$$\mathbb{E}_\xi\left[\mathbb{1}\{\xi \in [\zeta_i - 2\kappa, \zeta_i]\} \cdot (\xi + 2\kappa - \zeta_i(c'))\right]$$
$$\geq \mathbb{E}_\xi\left[\mathbb{1}\{\xi \in [\zeta_i - 2\tilde{\kappa}, \zeta_i]\} \cdot (\xi + 2\kappa - \zeta_i(c'))\right]$$
$$\geq \int_{\zeta_i(c')-2\tilde{\kappa}}^{\zeta_i(c')} \frac{\alpha e^{-\frac{(\xi - \bar{\xi})^2}{2\sigma^2}}\mathbb{1}\{\|c'\| \leq \frac{D}{m_i}\}}{\sqrt{2\pi\sigma^2}} \cdot (\xi + 2\kappa - \zeta_i(c'))\mathrm{d}\xi,$$

for all $\tilde{\kappa} \in [0, \kappa]$. Therefore, it holds that

$$\mathbb{E}_\xi\left[(c + 2\Delta)^T(w^*(c) - w^*(c + 2\Delta))\right]$$
$$\geq \sum_{i=1}^{k-1}(y_i - y_{i+1})\tilde{\kappa}\kappa \cdot \frac{\alpha e^{-\frac{3(\zeta_i(c')^2 + \tilde{\kappa}^2 + \bar{\xi}^2)}{2\sigma^2}}\mathbb{1}\{\|c'\| \leq \frac{D}{m_i}\}}{2\sqrt{2\pi\sigma^2}}$$
$$= \frac{\alpha\tilde{\kappa}\kappa e^{-\frac{3(\tilde{\kappa}^2 + \bar{\xi}^2)}{2\sigma^2}}}{2} \cdot \sum_{i=1}^{k-1}\frac{e^{-\frac{3\zeta_i^2(c')}{2\sigma^2}}\mathbb{1}\{\|c'\| \leq \frac{D}{m_i}\}}{\sqrt{2\pi\sigma^2}}(y_i - y_{i+1}).$$

$\square$

Lemma B.5 provide a lower bound of the conditional SPO+ risk when the distribution of $c = (c', \epsilon)$ is well behaved.

**Lemma B.5.** *Let $\bar{c}' \in \mathbb{R}^{d-1}$ be a fixed vector and $\bar{\xi} \in \mathbb{R}$, $\sigma > 0$ be fixed scalars. Let $c' \in \mathbb{R}^{d-1}$ be a random vector satisfying $\mathbb{P}(c') \geq \mathcal{N}(\bar{c}', \sigma^2 I_{d-1})$ for all $\|c'\|_2^2 \leq 2D^2$, and let $\xi \in \mathbb{R}$ be a random variable satisfying $\mathbb{P}(\xi|c') \geq \alpha \cdot \mathcal{N}(\bar{\xi}, \sigma^2)$ for all $\xi \in [-\sqrt{2D^2 - \|c'\|^2}, \sqrt{2D^2 - \|c'\|^2}]$. Define $\Xi_S := (1 + \frac{2\sqrt{3}D_S}{d_S})^{1-d}$. Suppose $\Delta = \kappa \cdot e_d$ for some $\kappa > 0$, then for all $\tilde{\kappa} \in [0, \kappa]$, it holds that*

$$\mathbb{E}_{c', \xi}\left[(c + 2\Delta)^T(w^*(c) - w^*(c + 2\Delta))\right] \geq \frac{\alpha\tilde{\kappa}\kappa e^{-\frac{3\tilde{\kappa}^2 + 3\bar{\xi}^2 + \|\bar{c}'\|_2^2}{2\sigma^2}}}{4\sqrt{2\pi\sigma^2}} \cdot \frac{\gamma(\frac{d-1}{2}, D^2)}{\Gamma(\frac{d-1}{2})} \cdot \Xi_S(y_- - y_+).$$

*Proof of Lemma B.5.* By result in Lemma B.4, it holds that

$$\mathbb{E}_{c', \xi}\left[(c + 2\Delta)^T(w^*(c) - w^*(c + 2\Delta))\right]$$
$$\geq \frac{\alpha\tilde{\kappa}\kappa e^{-\frac{3(\tilde{\kappa}^2 + \bar{\xi}^2)}{2\sigma^2}}}{2} \cdot \mathbb{E}_{c'}\left[\sum_{i=1}^{k(c')-1}\frac{e^{-\frac{3\zeta_i^2(c')}{2\sigma^2}}\mathbb{1}\{\|c'\| \leq \frac{D}{m_i}\}}{\sqrt{2\pi\sigma^2}}(y_i(c') - y_{i+1}(c'))\right].$$

For any $c' \in \mathbb{R}^{d-1}$, let $r = \|c'\|_2$ and $\hat{c}' = \frac{c'}{r}$. We know that $k(c') = k(\hat{c}')$, $\zeta_i(c') = r\zeta_i(\hat{c}')$, $w_i(c') = w_i(\hat{c}')$, and $y_i(c') = y_i(\hat{c}')$. Then we have

$$\mathbb{E}_{c'}\left[\sum_{i=1}^{k(c')-1} \frac{e^{-\frac{3\zeta_i^2(c')}{2\sigma^2}}}{\sqrt{2\pi\sigma^2}}(y_i(c') - y_{i+1}(c'))\right]$$

$$= \int_{\mathbb{S}^{d-2}} \int_0^\infty \sum_{i=1}^{k(\hat{c}')-1} \frac{e^{-\frac{3r^2\zeta_i^2(\hat{c}')}{2\sigma^2}}\mathbb{1}\{r\leq\frac{D}{m_i}\}}{\sqrt{2\pi\sigma^2}}(y_i(\hat{c}') - y_{i+1}(\hat{c}'))r^{d-2}\mathbb{P}_{c'}(r\hat{c}')\mathrm{d}r\mathrm{d}\hat{c}',$$

where $\mathbb{S}^{d-2} = \{\hat{c}' \in \mathbb{R}^{d-1} : \|\hat{c}'\|_2 = 1$. For fixed $\hat{c}' \in \mathbb{S}^{d-2}$ with $\hat{c}'^T\bar{c}' \geq 0$ and $i \in \{1,\ldots,k(\hat{c}') - 1\}$, we have

$$\int_0^{\frac{D}{m_i}} \frac{e^{-\frac{3r^2\zeta_i^2(\hat{c}')}{2\sigma^2}}}{\sqrt{2\pi\sigma}}r^{d-2}\mathbb{P}_{c'}(r\hat{c}')\mathrm{d}r = \int_0^{\frac{D}{m_i}} \frac{e^{-\frac{3r^2\zeta_i^2(\hat{c}')}{2\sigma^2}}}{\sqrt{2\pi\sigma^2}} \cdot \frac{e^{-\frac{\|r\hat{c}'-\bar{c}'\|_2^2}{2\sigma^2}}}{(2\pi\sigma^2)^{\frac{d-1}{2}}} \cdot r^{d-2}\mathrm{d}r$$

$$\geq \int_0^{\frac{D}{m_i}} \frac{e^{-\frac{3r^2\zeta_i^2(\hat{c}')}{2\sigma^2}}}{\sqrt{2\pi\sigma^2}} \cdot \frac{e^{-\frac{r^2+\|\bar{c}'\|_2^2}{2\sigma^2}}}{(2\pi\sigma^2)^{\frac{d-1}{2}}} \cdot r^{d-2}\mathrm{d}r$$

$$= \frac{e^{-\frac{\|\bar{c}'\|_2^2}{2\sigma^2}}}{\sqrt{2\pi\sigma^2}} \cdot \frac{\gamma(\frac{d-1}{2}, \frac{D^2(1+3\zeta_i^2(\hat{c}'))}{m_i^2})}{2\pi^{\frac{d-1}{2}}} \cdot (1 + 3\zeta_i^2(\hat{c}'))^{-\frac{d-1}{2}}$$

$$\geq \frac{e^{-\frac{\|\bar{c}'\|_2^2}{2\sigma^2}}}{\sqrt{2\pi\sigma^2}} \cdot \frac{\gamma(\frac{d-1}{2}, D^2)}{2\pi^{\frac{d-1}{2}}} \cdot (1 + 3\zeta_i^2(\hat{c}'))^{-\frac{d-1}{2}},$$

where $\gamma(\cdot, \cdot)$ is the lower incomplete Gamma function. By Lemma B.3, it holds that

$$\sum_{i=1}^{k(\hat{c}')-1} (1 + 3\zeta_i^2(\hat{c}'))^{-\frac{d-1}{2}}(y_i(\hat{c}') - y_{i+1}(\hat{c}')) \geq \Xi_{S,\hat{c}'} \cdot (y_- - y_+). \tag{5}$$

Therefore, it holds that

$$\mathbb{E}_{c'}\left[\sum_{i=1}^{k(c')-1} \frac{e^{-\frac{3\zeta_i^2(c')}{2\sigma^2}}}{\sqrt{2\pi\sigma^2}}(y_i(c') - y_{i+1}(c'))\right]$$

$$\geq \int_{\mathbb{S}^{d-2}} \mathbb{1}\{\hat{c}'^T\bar{c}' \geq 0\} \cdot \frac{e^{-\frac{\|\bar{c}'\|_2^2}{2\sigma^2}}}{\sqrt{2\pi\sigma^2}} \cdot \frac{\gamma(\frac{d-1}{2}, D^2)}{2\pi^{\frac{d-1}{2}}} \cdot \Xi_{S,c'}(y_- - y_+)\mathrm{d}\hat{c}'$$

$$\geq \frac{e^{-\frac{\|\bar{c}'\|_2^2}{2\sigma^2}}}{2\sqrt{2\pi\sigma^2}} \cdot \frac{\gamma(\frac{d-1}{2}, D^2)}{\Gamma(\frac{d-1}{2})} \cdot \Xi_S(y_- - y_+),$$

and finally we get

$$\mathbb{E}_{c',\xi}\left[(c + 2\Delta)^T(w^*(c) - w^*(c + 2\Delta))\right]$$

$$\geq \frac{\alpha\tilde{\kappa}\kappa e^{-\frac{3(\tilde{\kappa}^2+\xi^2)}{2\sigma^2}}}{2} \cdot \frac{e^{-\frac{\|\bar{c}'\|_2^2}{2\sigma^2}}}{2\sqrt{2\pi\sigma^2}} \cdot \frac{\gamma(\frac{d-1}{2}, D^2)}{\Gamma(\frac{d-1}{2})} \cdot \Xi_S(y_- - y_+)$$

$$= \frac{\alpha\tilde{\kappa}\kappa e^{-\frac{3\tilde{\kappa}^2+3\bar{\xi}^2+\|\bar{c}'\|_2^2}{2\sigma^2}}}{4\sqrt{2\pi\sigma^2}} \cdot \frac{\gamma(\frac{d-1}{2}, D^2)}{\Gamma(\frac{d-1}{2})} \cdot \Xi_S(y_- - y_+).$$

$\square$

Now we present a general version of Theorem 3.1. For given parameters $M \geq 1$ and $\alpha, \beta, D > 0$, define $\mathcal{P}_{M,\alpha,\beta,D} := \{\mathbb{P} \in \mathcal{P}_{\text{cont, symm}} : \text{for all } x \in \mathcal{X} \text{ with } \bar{c} = \mathbb{E}[c|x], \text{ there exists } \sigma \in [0, \min\{D, M\}] \text{ satisfying } \|\bar{c}\|_2 \leq \beta\sigma \text{ and } \mathbb{P}(c|x) \geq \alpha \cdot \mathcal{N}(\bar{c}, \sigma^2 I) \text{ for all } c \in \mathbb{R}^d \text{ satisfying } \|c\|_2^2 \leq 2D^2\}$. By introducing the constant $D$, we no longer require the conditional distribution $\mathbb{P}(c|x)$ be lower bounded by a normal distribution on the entire vector space $\mathbb{R}^d$. Instead, we only need $\mathbb{P}(c|x)$ has a lower bound on a bounded $\ell_2$-ball.

**Theorem B.1** (A general version of Theorem 3.1). *Suppose the feasible region $S$ is a polyhedron and define $\Xi_S := (1 + \frac{2\sqrt{3}D_S}{d_S})^{1-d}$. Then the calibration function of the SPO+ loss satisfies*

$$\hat{\delta}_{\ell_{\text{SPO+}}}(\epsilon; \mathcal{P}_{M,\alpha,\beta,D}) \geq \frac{\alpha \Xi_S \gamma(\frac{d-1}{2}, D^2)}{4\sqrt{2\pi}e^{\frac{3(1+\beta^2)}{2}}\Gamma(\frac{d-1}{2})} \cdot \min\left\{\frac{\epsilon^2}{D_S M}, \epsilon\right\} \quad \text{for all } \epsilon > 0. \tag{6}$$

*Additionally, when $D = \infty$, we have $\gamma(\frac{d-1}{2}, D^2) = \Gamma(\frac{d-1}{2})$ and therefore*

$$\hat{\delta}_{\ell_{\text{SPO+}}}(\epsilon; \mathcal{P}_{M,\alpha,\beta}) \geq \frac{\alpha \Xi_S}{4\sqrt{2\pi}e^{\frac{3(1+\beta^2)}{2}}} \cdot \min\left\{\frac{\epsilon^2}{D_S M}, \epsilon\right\} \quad \text{for all } \epsilon > 0.$$

*Proof of Theorem B.1.* Without loss of generality, we assume $d_S > 0$. Otherwise, the constant $\Xi_S$ will be zero and (6) will be a trivial bound. Let $\kappa = \|\Delta\|_2$ and $A \in \mathbb{R}^{d \times d}$ be an orthogonal matrix such that $A^T \Delta = \kappa \cdot e_d$ for $e_d = (0, \ldots, 0, 1)^T$. We implement a change of basis and let the new basis be $A = (a_1, \ldots, a_d)$. With a slight abuse of notation, we keep the notation the same after the change of basis, for example, now the vector $\Delta$ equals to $\kappa \cdot e_d$. Since the excess SPO risk of $\hat{c} = \bar{c} + \Delta$ is at least $\epsilon$, we have $\kappa(y_- - y_+) \geq \epsilon$. Let $\tilde{\kappa} = \min\{\kappa, \sigma\}$. Then it holds that $\tilde{\kappa}\exp(-\frac{3\tilde{\kappa}^2}{2\sigma^2}) \geq \min\{\kappa, \sigma\} \cdot \exp(-\frac{3}{2})$. By Lemma B.5, we know that

$$R_{\text{SPO}}(\hat{c}) = \mathbb{E}_c\left[(c + 2\Delta)^T(w^*(c) - w^*(c + 2\Delta))\right]$$

$$\geq \frac{\tilde{\kappa}\kappa e^{-\frac{3\tilde{\kappa}^2 + 3\bar{\xi}^2 + \|\bar{c}'\|_2^2}{2\sigma^2}}\gamma(\frac{d-1}{2}, D^2)}{4\sqrt{2\pi\sigma^2}\Gamma(\frac{d-1}{2})} \cdot \Xi_S(y_- - y_+),$$

where $\bar{c}'$ is the first $(d-1)$ elements of $\bar{c}$, $\bar{\xi}$ is the last element of $\bar{c}$, and $3\bar{\xi}^2 + \|\bar{c}'\|_2^2 \leq 3\|\bar{c}\|_2^2 = 3\alpha^2\sigma^2$. Then we can conclude that

$$R_{\text{SPO+}}(\hat{c}) - R_{\text{SPO+}}^* \geq \frac{\alpha \Xi_S \gamma(\frac{d-1}{2}, D^2) \cdot \epsilon}{4\sqrt{2\pi}e^{\frac{3(1+\beta^2)}{2}}\Gamma(\frac{d-1}{2})} \cdot \min\left\{\frac{\kappa}{\sigma}, 1\right\}.$$

Furthermore, since $\frac{\kappa}{\sigma} \geq \frac{\kappa}{M} \geq \frac{\epsilon}{(y_- - y_+)M} \geq \frac{\epsilon}{D_S M}$, we have

$$R_{\text{SPO+}}(\hat{c}) - R_{\text{SPO+}}^* \geq \frac{\alpha \Xi_S \gamma(\frac{d-1}{2}, D^2)}{4\sqrt{2\pi}e^{\frac{3(1+\beta^2)}{2}}\Gamma(\frac{d-1}{2})} \cdot \min\left\{\frac{\epsilon^2}{D_S M}, \epsilon\right\}.$$

$\square$

## B.4 Tightness of the lower bound in Theorem B.1.

Herein we provide an example to show the tightness of the lower bound in Theorem B.1.

*Example* 2. For any given $\epsilon > 0$, we consider the conditional distribution $\mathbb{P}(c|x) = \mathcal{N}(-\epsilon' \cdot e_d, \sigma^2 I_d)$ for some constants $\epsilon', \sigma > 0$ to be determined. For some $a, b > 0$, let the feasible region $S$ be $S = \text{conv}(\{w \in \mathbb{R}^d : \|w_{1:(d-1)}\|_2 = a, w_d = 0\} \cup \{\pm b \cdot e_d\})$. Although $S$ is not polyhedral, it can be considered as a limiting case of a polyhedron and the argument easily extends, with minor complications, to the case where the sphere is replaced by an $(d-1)$-gon for $d$ sufficiently large. Let $\hat{c} = \epsilon' \cdot e_d$, we have $\mathbb{E}\left[\ell_{\text{SPO}}(\hat{c}, c)|x\right] - \mathbb{E}\left[\ell_{\text{SPO}}(\bar{c}, c)|x\right] = 2b\epsilon'$. Also, for the excess conditional SPO+ risk we have

$$\mathbb{E}\left[\ell_{\text{SPO+}}(\hat{c}, c)|x\right] - \mathbb{E}\left[\ell_{\text{SPO+}}(\bar{c}, c)|x\right] \to \int_{\mathbb{R}^{d-1}} \prod_{j=1}^{d-1} \frac{e^{-\frac{c_j^2}{2\sigma^2}}}{\sqrt{2\pi\sigma^2}} \cdot \frac{e^{-\frac{a^2\sum_{j=1}^{d-1}c_j^2}{2b^2\sigma^2}}}{\sqrt{2\pi\sigma^2}} \cdot \frac{\epsilon'^2}{2} \cdot dc_1 \ldots dc_{d-1}$$

$$= \frac{\epsilon'^2}{2\sqrt{2\pi\sigma^2}} \prod_{j=1}^{d-1} \int_{\mathbb{R}} \frac{e^{-\frac{c_j^2}{2\sigma^2}} \cdot e^{-\frac{a^2 c_j^2}{2b^2\sigma^2}}}{\sqrt{2\pi\sigma^2}} dc_j$$

$$= \frac{\epsilon'^2}{2\sqrt{2\pi\sigma^2}} \cdot \left(\frac{b^2}{a^2 + b^2}\right)^{(d-1)/2},$$

when $\epsilon' \to 0$. Therefore, let $\epsilon' = \frac{\epsilon}{2b}$, we have $\mathbb{E}\left[\ell_{\text{SPO}}(\hat{c}, c)|x\right] - \mathbb{E}\left[\ell_{\text{SPO}}(\bar{c}, c)|x\right] = \epsilon$ and

$$\mathbb{E}\left[\ell_{\text{SPO+}}(\hat{c}, c)|x\right] - \mathbb{E}\left[\ell_{\text{SPO+}}(\bar{c}, c)|x\right] = \frac{\epsilon^2}{8\sqrt{2\pi\sigma^2}b^2} \cdot \left(\frac{b^2}{a^2+b^2}\right)^{(d-1)/2}$$

$$\leq \frac{1}{8\sqrt{2\pi}} \cdot \left(\frac{D_S}{d_S}\right)^{1-d} \cdot \frac{\epsilon^2}{\sigma},$$

for some $b$ large enough, and therefore the lower bound in Theorem B.1 is tight up to a constant.

## C  Proofs and other technical details for Section 4

### C.1  Extension of Theorem C.1 and Lemma C.1 and C.2.

Let us first present a more general version of Assumption C.1 that allows the domain of the strongly convex function to be a subset of $\mathbb{R}^d$. In particular, we define the domain set $T \subseteq \mathbb{R}^d$ by $T := \{w \in \mathbb{R}^d : h_i^T w = s_i \ \forall i \in [m_1], t_j(w) < 0 \ \forall j \in [m_2]\}$, where $h_i \in \mathbb{R}^{\bar{d}}$ and $s_i \in \mathbb{R}$ for $i \in [m_1]$, and $t_j(\cdot) : \mathbb{R}^d \to \mathbb{R}$ are convex functions for $j \in [m_2]$. Clearly, when $m_1 = m_2 = 0$, the set $T$ is the entire vector space $\mathbb{R}^d$. Also, let the closure of $T$ be $\bar{T} = \{w \in \mathbb{R}^d : h_i^T w = s_i \ \forall i \in [m_1], t_j(w) \leq 0 \ \forall j \in [m_2]\}$, and with a slight abuse of notation, let the (relative) boundary of $T$ be $\partial T := \bar{T} \backslash T$. For any function defined on $T$, we consider the (relative) lower limit be $\underline{\lim}_{w \to \partial T} = \inf_{\delta > 0} \sup_{w \in T : d(w, \partial T) \leq \delta} f(w)$, where the distance function $d(\cdot, \cdot)$ is defined as $d(w, \partial T) = \min_{w' \in \partial T} \|w - w'\|_2$.

**Assumption C.1** (Generalization of Assumption 4.1). *For a given norm $\| \cdot \|$, let $f : T \to \mathbb{R}$ be a $\mu$-strongly convex function on $T$ for some $\mu > 0$. Assume that the feasible region $S$ is defined by $S = \{w \in T : f(w) \leq r\}$ for some constant $r$ satisfying $\underline{\lim}_{w \to \partial T} f(w) > r > f_{\min} := \min_{w \in T} f(w)$. Additionally, assume that $f$ is $L$-smooth on $S$ for some $L \geq \mu$.*

Let $H$ denote the linear subspace defined by the linear combination of all $h_j$, namely $H = \text{span}(\{h_j\}_{j=1}^{m_2})$, and let $H^\perp$ denote its orthogonal complement, namely $H^\perp = \{w \in \mathbb{R}^d : h_j^T w = 0, \forall j \in [m_2]\}$. Also, for any $c \in \mathbb{R}^d$, let $\text{proj}_{H^\perp}(c)$ denote its projection onto $H^\perp$. Theorem C.1 provides an $O(\epsilon)$ lower bound of the calibration function of two different distribution classes, which include the multi-variate Gaussian, Laplace, and Cauchy distribution. Theorem 4.1 is a special instance of Theorem C.1 when $m_1 = m_2 = 0$ (and thus $H^\perp = \mathbb{R}^d$).

**Theorem C.1** (Generalization of Theorem 4.1). *Suppose that Assumption C.1 holds with respect to the norm $\| \cdot \|_A$ for some positive definite matrix $A$. Then, for any $\epsilon > 0$, it holds that $\hat{\delta}_{\ell_{\text{SPO+}}}(\epsilon; \mathcal{P}_{\beta, A}) \geq \frac{\mu^{9/2}}{4(1+\beta^2)L^{9/2}} \cdot \epsilon$ and $\hat{\delta}_{\ell_{\text{SPO+}}}(\epsilon; \mathcal{P}_{\alpha, \beta, A}) \geq \frac{\alpha \mu^{9/2}}{4(1+\beta^2)L^{9/2}} \cdot \epsilon$.*

Our analysis for the calibration function relies on the following two lemmas, which utilize the property of the feasible region to strengthen the "first-order optimality" and provide a "Lipschitz-like" continuity of the optimization oracle. We want to mention that some of the results in the following two lemmas generalize the results in El Balghiti et al. [2019] to the cases where the feasible region $S$ is defined on a subspace of $\mathbb{R}^d$ rather than an open set in $\mathbb{R}^d$. The following lemma provides both upper and lower bound of SPO-like loss.

**Lemma C.1** (Generalization of Lemma 4.1). *Suppose Assumption C.1 holds with respect to a generic norm $\| \cdot \|$. Then, for any $c_1, c_2 \in \mathbb{R}^d$, it holds that*

$$c_1^T(w - w^*(c_1)) \geq \frac{\mu}{2\sqrt{2L(r - f_{\min})}}\|\text{proj}_{H^\perp}(c_1)\|_*\|w - w^*(c_1)\|^2, \quad \forall w \in S, \qquad (7)$$

*and*

$$c_1^T(w^*(c_2) - w^*(c_1)) \leq \frac{L}{2\sqrt{2\mu(r - f_{\min})}}\|\text{proj}_{H^\perp}(c_1)\|_*\|w^*(c_1) - w^*(c_2)\|^2. \qquad (8)$$

The two constants are the same since Theorem 12 in Journée et al. [2010] showed that set $S$ is a $\frac{\mu}{\sqrt{2Lr}}$-strongly convex set. The following lemma provides a lower bound in the difference between two optimal decision based on the difference between the two normalized cost vector.

**Lemma C.2** (Generalization of Lemma 4.2). *Suppose that Assumption 4.1 holds with respect to a generic norm $\|\cdot\|$. Let $c_1, c_2 \in \mathbb{R}^d$ be such that $\text{proj}_{H^\perp}(c_1), \text{proj}_{H^\perp}(c_2) \neq 0$, then it holds that*

$$\|w^*(c_1) - w^*(c_2)\| \geq \frac{\sqrt{2\mu(r - f_{\min})}}{L} \cdot \left\| \frac{\text{proj}_{H^\perp}(c_1)}{\|\text{proj}_{H^\perp}(c_1)\|_*} - \frac{\text{proj}_{H^\perp}(c_2)}{\|\text{proj}_{H^\perp}(c_2)\|_*} \right\|_*,$$

*and*

$$\|w^*(c_1) - w^*(c_2)\| \leq \frac{\sqrt{2L(r - f_{\min})}}{\mu} \cdot \left\| \frac{\text{proj}_{H^\perp}(c_1)}{\|\text{proj}_{H^\perp}(c_1)\|_*} - \frac{\text{proj}_{H^\perp}(c_2)}{\|\text{proj}_{H^\perp}(c_2)\|_*} \right\|_*.$$

## C.2  Proofs and useful lemmas

From now on, for any vector $c \in \mathbb{R}^d$, we will use $\tilde{c}$ to represent the projection $\text{proj}_{H^\perp}(c)$ for simplicity. Likewise, when $c = \nabla f(w)$ we shorten this notation even further to $\tilde{\nabla} f(w)$.

First we provide some useful properties in the following lemma.

**Lemma C.3.** *If $f(\cdot)$ is $\mu$-strongly convex on $S$, the for all $w \in S$, it holds that*

$$\|\tilde{\nabla} f(w)\|_*^2 \geq \sqrt{2\mu(f(w) - f_{\min})}.$$

*Proof.* First, for all $c \in \mathbb{R}^d$ and $w, w' \in S$, it holds that

$$c^T(w - w') - \tilde{c}^T(w - w') = (c - \tilde{c})^T(w - w') = \sum_{j=1}^{m_2} \alpha_j h_j(w - w') = 0.$$

Since $f(\cdot)$ is $\mu$-strongly convex, it holds that $f(w') \geq f(x) + \nabla f(w)^T(w' - w) + \frac{\mu}{2}\|w' - w\|^2$ for all $w' \in S$. Therefore, it holds that

$$
\begin{aligned}
\inf_{w' \in S} f(w') &\geq \inf_{w' \in S} \left\{ f(w) + \nabla f(w)^T(w' - w) + \frac{\mu}{2}\|w' - w\|^2 \right\} \\
&= \inf_{w' \in S} \left\{ f(w) + \tilde{\nabla} f(w)^T(w' - w) + \frac{\mu}{2}\|w' - w\|^2 \right\} \\
&\geq \inf_{w' \in \mathbb{R}^d} \left\{ f(w) + \tilde{\nabla} f(w)^T(w' - w) + \frac{\mu}{2}\|w' - w\|^2 \right\} \\
&= f(w) - \frac{1}{2\mu}\|\tilde{\nabla} f(w)\|_*^2.
\end{aligned}
$$

$\square$

**Lemma C.4.** *If $f(\cdot)$ is $L$-smooth on $S$, then for all $w \in S$, it holds that*

$$\|\tilde{\nabla} f(w)\|_*^2 \leq \sqrt{2L(f(w) - f_{\min})}.$$

*Proof.* If $\tilde{\nabla} f(w) = 0$, then the statement holds. Otherwise, there exists $u \in \mathbb{R}^d$ such that $\|u\| = 1$ and $\tilde{\nabla} f(w)^T u = \|\tilde{\nabla} f(w)\|_*$. Let $v = \|\tilde{\nabla} f(w)\|_* u$, we have $\|v\| = \|\tilde{\nabla} f(w)\|_*$ and $\tilde{\nabla} f(w)^T v = \|\tilde{\nabla} f(w)\|_*^2$. Let

$$\alpha = \sup \alpha', \quad \text{s.t. } f(w - \alpha' \tilde{v}) \leq r.$$

Since $g_i(\cdot)$ is continuous and $g_i(w) < 0$ for all $i \in [m_1]$, we have $\alpha > 0$ and since $f(\cdot)$ is continuous, we have $f(w - \alpha \tilde{v}) = r$. Since $f(\cdot)$ is $L$-smooth on $S$, it holds that

$$
\begin{aligned}
f(w - \alpha \tilde{v}) &\leq f(w) - \alpha \nabla f(w)^T \tilde{v} + \frac{\alpha^2 L}{2}\|\tilde{v}\|^2 = f(w) - \alpha \tilde{\nabla} f(w)^T \tilde{v} + \frac{\alpha^2 L}{2}\|\tilde{v}\|^2 \\
&= f(w) - \alpha \tilde{\nabla} f(w)^T v + \frac{\alpha^2 L}{2}\|\tilde{v}\|^2 \leq f(w) - \alpha \tilde{\nabla} f(w)^T v + \frac{\alpha^2 L}{2}\|v\|^2 \\
&= f(w) - \frac{2\alpha - \alpha^2 L}{2}\|\tilde{\nabla} f(w)\|_*^2.
\end{aligned}
$$

Therefore, we have $2\alpha - \alpha^2 L \leq 0$. Moreover, since $\alpha > 0$, then it holds that $\alpha \geq \frac{2}{L}$. Now we know that $w - \frac{\tilde{v}}{L} \in S$, and

$$f_{\min} \leq f\left(w - \frac{\tilde{v}}{L}\right) \leq f(w) - \frac{1}{2L}\|\tilde{\nabla} f(w)\|_*^2.$$

Therefore, it holds that $\|\tilde{\nabla} f(w)\|_* \leq \sqrt{2L(f(w) - f_{\min})}$. $\qquad\square$

Now we provide the proofs of Lemma C.1 and C.2.

*Proof of Lemma C.1.* Let $w_1 = w^*(c_1)$ and $w_2 = w^*(c_2)$. Since $f(\cdot)$ is $\mu$-strongly convex on $S$, it holds that

$$f(w) - f(w_1) - \nabla f(w_1)^T(w - w_1) \geq \frac{\mu}{2}\|w - w_1\|^2.$$

Since the Slater condition holds, the KKT necessary condition indicates that there exists scalar $u \geq 0$ such that $\tilde{c}_1 + u\tilde{\nabla} f(w_1) = 0$ and $u(f(w_1) - r) = 0$. When $\tilde{c}_1 \neq 0$, we additionally have $f(w_1) = r$. Therefore, it holds that

$$c_1^T(w - w_1) = \tilde{c}_1^T(w - w_1) = u \cdot \left(-\tilde{\nabla} f(w_1)^T(w - w_1)\right) = u \cdot \left(-\nabla f(w_1)^T(w - w_1)\right)$$

$$\geq u \cdot \left(f(w_1) - f(w) + \frac{\mu}{2}\|w - w_1\|^2\right) \geq \frac{u\mu}{2}\|w - w_1\|^2,$$

where the last inequality holds since $f(w_1) = r \geq f(w)$. Therefore, it holds that

$$c_1^T(w - w_1) \geq \frac{\mu\|\tilde{c}_1\|_*\|w - w_1\|^2}{2\|\tilde{\nabla} f(w_1)\|_*}. \tag{9}$$

Since $f(\cdot)$ is $L$-smooth on $S$, it holds that $\|\tilde{\nabla} f(w_1)\|_* \leq \sqrt{2L(r - f_{\min})}$, and hence we have

$$\frac{\|\tilde{c}_1\|_*}{\|\tilde{\nabla} f(w_1)\|_*} \geq \frac{\|\tilde{c}_1\|_*}{\sqrt{2L(r - f_{\min})}}.$$

By applying the above inequality to (9), we can conclude that

$$c^T(w - w_1) \geq \frac{\mu}{2\sqrt{2L(r - f_{\min})}}\|\tilde{c}_1\|_*\|w - w_1\|^2.$$

On the other hand, it holds that

$$c_1^T(w_2 - w_1) = \tilde{c}_1^T(w_2 - w_1) = u \cdot \left(-\tilde{\nabla} f(w_1)^T(w_2 - w_1)\right) = u \cdot \left(-\nabla f(w_1)^T(w_2 - w_1)\right)$$

$$\leq u \cdot \left(f(w_1) - f(w_2) + \frac{L}{2}\|w_2 - w_1\|^2\right) = \frac{uL}{2}\|w - w_1\|^2,$$

where the last inequality holds since $f(w_1) = r = f(w_2)$. Therefore, it holds that

$$c_1^T(w_2 - w_1) \leq \frac{L\|\tilde{c}_1\|_*\|w_2 - w_1\|^2}{2\|\tilde{\nabla} f(w_1)\|_*}. \tag{10}$$

Since $f(\cdot)$ is $\mu$-strongly convex on $S$, it holds that $\|\tilde{\nabla} f(w_1)\|_* \geq \sqrt{2\mu(r - f_{\min})}$, and hence we have

$$\frac{\|\tilde{c}_1\|_*}{\|\tilde{\nabla} f(w_1)\|_*} \leq \frac{\|\tilde{c}_1\|_*}{\sqrt{2\mu(r - f_{\min})}}.$$

By applying the above inequality to (10), we can conclude that

$$c^T(w - w_1) \leq \frac{L}{2\sqrt{2\mu(r - f_{\min})}}\|\tilde{c}_1\|_*\|w_2 - w_1\|^2.$$

$\qquad\square$

*Proof of Lemma C.2.* Without loss of generality we assume $\|\tilde{c}_1\|_* = \|\tilde{c}_2\|_* = 1$. Let $w_1 = w^*(c_1)$ and $w_2 = w^*(c_2)$. By KKT condition there exists $u_1, u_2 > 0$ such that $\nabla f(w_i) = -u_i c_i$ and $f(w_i) = r$ for $i = 1, 2$. Also, since $f(\cdot)$ is $\mu$-strongly convex, it holds that

$$\|\tilde{\nabla} f(w_i)\|_* \geq \sqrt{2\mu(f(x_i) - f_{\min})} = \sqrt{2\mu(r - f_{\min})},$$

for $i = 1, 2$. Then, it holds that

$$\|\tilde{\nabla} f(w_1) - \tilde{\nabla} f(w_2)\|_* \geq \min_{u_1', u_2' \geq \sqrt{2\mu(r - f_{\min})}} \|u_1' \tilde{c}_1 - u_2' \tilde{c}_2\|_* = \sqrt{2\mu(r - f_{\min})} \cdot \|\tilde{c}_1 - \tilde{c}_2\|_*.$$

Moreover, since $f(\cdot)$ is $L$-smooth, it holds that

$$\|w_1 - w_2\| \geq \frac{1}{L} \cdot \|\nabla f(w_1) - \nabla f(w_2)\|_* \geq \frac{1}{L} \cdot \|\tilde{\nabla} f(w_1) - \tilde{\nabla} f(w_2)\|_* \geq \frac{\sqrt{2\mu r}}{L} \cdot \|\tilde{c}_1 - \tilde{c}_2\|_*.$$

$\square$

In the rest part of this section, without loss of generality we assume $f_{\min} = 0$. Also, since $w * (c) = w^*(\tilde{c})$ and $c^T(w^*(c') - w^*(c)) = \tilde{c}^T(w^*(c') - w^*(c))$ for all $c, c' \in \mathbb{R}^d$, we will ignore the ˜ notation and assume all $c, c' \in H^\perp$. In Theorem C.2 we provide a lower bound of an SPO-like loss.

**Theorem C.2.** *When $c \neq 0$ and $c + 2\Delta \neq 0$, it holds that*

$$(c + 2\Delta)^T(w^*(c) - w^*(c + 2\Delta)) \geq \frac{\mu^2 r^{1/2}}{2^{1/2} L^{5/2}} \cdot \|c + 2\Delta\|_* \cdot \left\| \frac{c}{\|c\|_*} - \frac{c + 2\Delta}{\|c + 2\Delta\|_*} \right\|_*^2.$$

*When the norm we consider is $A$-norm defined by $\|x\|_A = \sqrt{x^T A x}$ for some positive definite matrix $A$, additionally we have*

$$(c + 2\Delta)^T(w^*(c) - w^*(c + 2\Delta)) \geq \frac{\mu^2 r^{1/2}}{2^{1/2} L^{5/2}} \left( \|c + 2\Delta\|_{A^{-1}} - \frac{c^T A^{-1}(c + 2\Delta)}{\|c\|_{A^{-1}}} \right).$$

*Moreover, if $\mathbb{P}(c = 0) = \mathbb{P}(c = -2\Delta) = 0$, it holds that*

$$\ell_{\text{SPO+}}(\Delta) \geq \frac{\mu^2 r^{1/2}}{2^{1/2} L^{5/2}} \cdot \mathbb{E}_c \left[ \|c + 2\Delta\|_{A^{-1}} - \frac{c^T A^{-1}(c + 2\Delta)}{\|c\|_{A^{-1}}} \right].$$

*Proof of Theorem C.2.* Apply $c_1 = c$ and $c_2 = c + 2\Delta$ to Lemma C.2, we have

$$\|w^*(c) - w^*(c + 2\Delta)\| \geq \sqrt{2\mu r} \cdot \left\| \frac{c}{\|c\|_*} - \frac{c + 2\Delta}{\|c + 2\Delta\|_*} \right\|_*.$$

By applying the above inequality to (7) we have

$$(c + 2\Delta)^T(w^*(c) - w^*(c + 2\Delta)) \geq \frac{\mu}{2\sqrt{2Lr}} \cdot \|c + 2\Delta\|_* \cdot \|w^*(c) - w^*(c + 2\Delta)\|^2$$

$$\geq \frac{\mu}{2\sqrt{2Lr}} \cdot \|c + 2\Delta\|_* \cdot \left( \frac{\sqrt{2\mu r}}{L} \cdot \left\| \frac{c}{\|c\|_*} - \frac{c + 2\Delta}{\|c + 2\Delta\|_*} \right\|_* \right)^2$$

$$= \frac{\mu^2 r^{1/2}}{2^{1/2} L^{5/2}} \cdot \|c + 2\Delta\|_* \cdot \left\| \frac{c}{\|c\|_*} - \frac{c + 2\Delta}{\|c + 2\Delta\|_*} \right\|_*^2.$$

When the norm we consider is $A$-norm, then it holds that

$$(c + 2\Delta)^T(w^*(c) - w^*(c + 2\Delta)) \geq \frac{\mu^2 r^{1/2}}{2^{1/2} L^{5/2}} \cdot \|c + 2\Delta\|_2 \cdot \left\| \frac{c}{\|c\|_{A^{-1}}} - \frac{c + 2\Delta}{\|c + 2\Delta\|_{A^{-1}}} \right\|_{A^{-1}}^2$$

$$= \frac{\mu^2 r^{1/2}}{2^{1/2} L^{5/2}} \left( \|c + 2\Delta\|_{A^{-1}} - \frac{c^T A^{-1}(c + 2\Delta)}{\|c\|_{A^{-1}}} \right).$$

Moreover, if $\mathbb{P}(c = 0) = \mathbb{P}(c = -2\Delta) = 0$, by taking the expectation of $c$ we get

$$\ell_{\text{SPO+}}(\Delta) \geq \frac{\mu^2 r^{1/2}}{2^{1/2} L^{5/2}} \cdot \mathbb{E}_c \left[ \|c + 2\Delta\|_{A^{-1}} - \frac{c^T A^{-1}(c + 2\Delta)}{\|c\|_{A^{-1}}} \right].$$

$\square$

The following lemma provides a necessary condition on $\Delta$ such that the excess SPO loss of $\hat{c} = \bar{c} + \Delta$ is at least $\epsilon$.

**Lemma C.5.** *Suppose the excess SPO loss of $\hat{c} = \bar{c} + \Delta$ is at least $\epsilon$, that is, $\bar{c}^T(w^*(\bar{c}+\Delta) - w^*(\bar{c})) \geq \epsilon$. Then it holds that*

$$\left\| \frac{\bar{c}}{\|\bar{c}\|_*} - \frac{\bar{c} + \Delta}{\|\bar{c} + \Delta\|_*} \right\|_*^2 \geq \frac{2^{1/2}\mu^{5/2}\epsilon}{L^2 r^{1/2}\|\bar{c}\|_*}.$$

*When the norm we consider is A-norm defined by $\|x\|_A = \sqrt{x^T A x}$ for some positive definite matrix A, additionally we have*

$$1 - \frac{\bar{c}^T A^{-1}(\bar{c} + \Delta)}{\|\bar{c}\|_{A^{-1}} \cdot \|\bar{c} + \Delta\|_{A^{-1}}} \geq \frac{\mu^{5/2}}{2^{1/2}L^2 r^{1/2}\|\bar{c}\|_{A^{-1}}} \cdot \epsilon.$$

*Proof of Lemma C.5.* In Lemma C.1 we show that

$$c_1^T(w^*(c_2) - w^*(c_1)) \leq \frac{L}{2\sqrt{2\mu r}}\|c_1\|_* \|w^*(c_1) - w^*(c_2)\|^2.$$

Let $c_1 = \bar{c}$ and $c_2 = \hat{c}$, it holds that

$$\|w^*(\bar{c}) - w^*(\bar{c} + \Delta)\|^2 \geq \frac{2\sqrt{2\mu r}}{L\|\bar{c}\|_*} \cdot \bar{c}^T(w^*(\bar{c} + \Delta) - w^*(\bar{c})) \geq \frac{2\sqrt{2\mu r}\epsilon}{L\|\bar{c}\|_*}.$$

Theorem 3 in El Balghiti et al. [2019] shows that for $c_1, c_2 \in \mathbb{R}^d$, it holds that

$$\|c_1 - c_2\|_* \geq \frac{\mu}{\sqrt{2Lr}} \cdot \min\{\|c_1\|_*, \|c_2\|_*\} \cdot \|w^*(c_1) - w^*(c_2)\|.$$

By applying $c_1 = \frac{\bar{c}}{\|\bar{c}\|_*}$ and $c_2 = \frac{\bar{c}+\Delta}{\|\bar{c}+\Delta\|_*}$, we have

$$\left\| \frac{\bar{c}}{\|\bar{c}\|_*} - \frac{\bar{c} + \Delta}{\|\bar{c} + \Delta\|_*} \right\|^2 \geq \frac{\mu^2}{2Lr} \cdot \left\| w^*\left(\frac{\bar{c}}{\|\bar{c}\|_*}\right) - w^*\left(\frac{\bar{c} + \Delta}{\|\bar{c} + \Delta\|_*}\right) \right\|^2$$

$$= \frac{\mu^2}{2Lr} \cdot \|w^*(\bar{c}) - w^*(\bar{c} + \Delta)\|_*^2 \geq \frac{2^{1/2}\mu^{5/2}\epsilon}{L^2 r^{1/2}\|\bar{c}\|_*}.$$

When the norm we consider is 2-norm, it holds that

$$1 - \frac{\bar{c}^T A^{-1}(\bar{c} + \Delta)}{\|\bar{c}\|_{A^{-1}} \cdot \|\bar{c} + \Delta\|_{A^{-1}}} = \frac{1}{2}\left\| \frac{\bar{c}}{\|\bar{c}\|_{A^{-1}}} - \frac{\bar{c} + \Delta}{\|\bar{c} + \Delta\|_{A^{-1}}} \right\|_{A^{-1}}^2 \geq \frac{\mu^{5/2}\epsilon}{2^{1/2}L^2 r^{1/2}\|\bar{c}\|_{A^{-1}}}.$$

$\square$

From Theorem C.2 and Lemma C.5, we know that $\ell_{\text{SPO+}}(c, \Delta)$ have a lower bound $C_1(\mu, L, r) \cdot \underline{\ell}_{\text{SPO+}}(c, \Delta)$, where $C_1(\mu, L, r) = \frac{\mu^2 r^{1/2}}{2^{1/2}L^{5/2}}$ and

$$\underline{\ell}_{\text{SPO+}}(c, \Delta) = \|c + 2\Delta\|_{A^{-1}} - \frac{c^T A^{-1}(c + 2\Delta)}{\|c\|_{A^{-1}}}.$$

Moreover, the excess SPO risk of $\hat{c} = \bar{c} + \Delta$ is at least $\epsilon$ implies that $\overline{R}_{\text{SPO}}(\Delta) \geq C_2(\mu, L, r) \cdot \epsilon$ where $C_2(\mu, L, r) = \frac{\mu^{5/2}}{2^{1/2}L^2 r^{1/2}}$ and

$$\overline{R}_{\text{SPO}}(\Delta) = \|\bar{c}\|_{A^{-1}} - \frac{\bar{c}^T A^{-1}(\bar{c} + \Delta)}{\|\bar{c} + \Delta\|_{A^{-1}}}.$$

Let $\underline{R}_{\text{SPO+}}(\Delta) = \mathbb{E}_c[\underline{\ell}_{\text{SPO+}}(c, \Delta)]$. We know that the calibration function $\delta(\epsilon)$ has a lower bound $\delta'(\epsilon)$ which defined as

$$\begin{aligned} \delta'(\epsilon) := \min_{\Delta} \quad & C_1(\mu, L, r) \cdot \underline{R}_{\text{SPO+}}(\Delta) \\ \text{s.t.} \quad & \overline{R}_{\text{SPO}}(\Delta) \geq C_2(\mu, L, r) \cdot \epsilon. \end{aligned} \tag{11}$$

Here we first provide two properties of random variable $c$ when $\mathbb{P} \in \mathcal{P}_{\text{rot symm}}$.

**Proposition C.1.** *Suppose* $\mathbb{P} \in \mathcal{P}_{\text{rot symm}}$. *If* $\|\bar{c} + \zeta\|_{A^{-1}} = \|\bar{c}\|_{A^{-1}}$ *for some* $\zeta \in \mathbb{R}^p$, *it holds that*

$$\mathbb{E}_c\left[\|c + \zeta\|_{A^{-1}}\right] = \mathbb{E}_c\left[\|c\|_{A^{-1}}\right].$$

**Proposition C.2.** *Suppose* $\mathbb{P} \in \mathcal{P}_{\text{rot symm}}$. *When* $d \geq 2$, *for any constant* $t \geq 0$, *it holds that*

$$\mathbb{E}_c\left[\left.\frac{\bar{c}^T A^{-1} c}{\|c\|_{A^{-1}}}\right| \|c - \bar{c}\|_{A^{-1}} = t\right] \geq \frac{\|c\|_{A^{-1}}^2 \min\{\|c\|_{A^{-1}}, t\}}{t^2 + \|\bar{c}\|_{A^{-1}} t}.$$

*Proof of Proposition C.2.* For simplicity we just assume $\|c - \bar{c}\|_{A^{-1}} = t$ from now on and ignore the conditional probability. Let $\omega = c - \bar{c}$. Since $p(c) = p(2\bar{c} - c)$, we have

$$\mathbb{E}_c\left[\frac{\bar{c}^T A^{-1} c}{\|c\|_{A^{-1}}}\right] = \frac{1}{2} \cdot \mathbb{E}_c\left[\frac{\bar{c}^T A^{-1} c}{\|c\|_{A^{-1}}} + \frac{\bar{c}^T A^{-1}(2\bar{c} - c)}{\|2\bar{c} - c\|_{A^{-1}}}\right]$$

$$= \frac{1}{2} \cdot \mathbb{E}_\omega\left[\frac{\bar{c}^T A^{-1}(\bar{c} + \omega)}{\|\bar{c} + \omega\|_{A^{-1}}} + \frac{\bar{c}^T A^{-1}(\bar{c} - \omega)}{\|\bar{c} - \omega\|_{A^{-1}}}\right].$$

By the fact that $\bar{c}^T A^{-1} \bar{c}(\|\bar{c} - w\|_{A^{-1}} + \|\bar{c} + w\|_{A^{-1}}) \geq 2\|\bar{c}\|_{A^{-1}}^2 \|w\|_{A^{-1}} \geq \bar{c}^T A^{-1} w(\|\bar{c} - w\|_{A^{-1}} - \|\bar{c} + w\|_{A^{-1}})$, it holds that $\bar{c}^T A^{-1}(\bar{c} + w)\|\bar{c} - w\|_{A^{-1}} + \bar{c}^T A^{-1}(\bar{c} - w)\|\bar{c} - w\|_{A^{-1}} \geq 0$ and hence

$$\frac{\bar{c}^T A^{-1}(\bar{c} + \omega)}{\|\bar{c} + \omega\|_{A^{-1}}} + \frac{\bar{c}^T A^{-1}(\bar{c} - \omega)}{\|\bar{c} - \omega\|_{A^{-1}}} \geq 0.$$

Therefore, we further get

$$\mathbb{E}_c\left[\frac{\bar{c}^T A^{-1} c}{\|c\|_{A^{-1}}}\right] \geq \frac{1}{2} \cdot \mathbb{E}_\omega\left[\left.\frac{\bar{c}^T A^{-1}(\bar{c} + \omega)}{\|\bar{c} + \omega\|_{A^{-1}}} + \frac{\bar{c}^T A^{-1}(\bar{c} - \omega)}{\|\bar{c} - \omega\|_{A^{-1}}}\right| \bar{c}^T \omega \in C\right] \cdot \mathbb{P}(\bar{c}^T \omega \in C),$$

where $C = [-\|\bar{c}\|_{A^{-1}}^2, \|\bar{c}\|_{A^{-1}}^2]$. For any $\omega$ such that $\bar{c}^T \omega \in C$, we have

$$\frac{\bar{c}^T A^{-1}(\bar{c} + \omega)}{\|\bar{c} + \omega\|_{A^{-1}}} + \frac{\bar{c}^T A^{-1}(\bar{c} - \omega)}{\|\bar{c} - \omega\|_{A^{-1}}} \geq \frac{\bar{c}^T A^{-1}(\bar{c} + \omega)}{\|\bar{c}\|_{A^{-1}} + \|\omega\|_{A^{-1}}} + \frac{\bar{c}^T A^{-1}(\bar{c} - \omega)}{\|\bar{c}\|_{A^{-1}} + \|\omega\|_{A^{-1}}}$$

$$= \frac{2\bar{c}^T A^{-1} \bar{c}}{\|\bar{c}\|_{A^{-1}} + \|\omega\|_{A^{-1}}}.$$

Also, when $d \geq 2$, we have $\mathbb{P}(\bar{c}^T \omega \in C) \geq \frac{\min\{\|\bar{c}\|, t\}}{t}$. Then we can conclude that

$$\mathbb{E}_c\left[\frac{\bar{c}^T A^{-1} c}{\|c\|_{A^{-1}}}\right] \geq \frac{\|\bar{c}\|_{A^{-1}}^2 \min\{\|\bar{c}\|_{A^{-1}}, t\}}{t^2 + \|\bar{c}\|_{A^{-1}} t}.$$

$\square$

By first-order necessary condition we know that $\Delta$ is an optimal solution to (11) only if

$$\nabla \underline{R}_{\text{SPO+}}(\Delta) - \alpha \nabla \overline{R}_{\text{SPO}}(\Delta) = 0 \tag{12}$$

for some $\alpha \geq 0$. Also, for any fixed $\Delta$, it holds that

$$\nabla \underline{R}_{\text{SPO+}}(\Delta) = \mathbb{E}_c\left[\frac{A^{-1}(c + 2\Delta)}{\|c + 2\Delta\|_{A^{-1}}} - \frac{A^{-1} c}{\|c\|_{A^{-1}}}\right],$$

and

$$\nabla \overline{R}_{\text{SPO}}(\Delta) = \frac{\bar{c}^T A^{-1}(\bar{c} + \Delta) \cdot A^{-1} \Delta - \Delta^T A^{-1}(\bar{c} + \Delta) \cdot A^{-1} \bar{c}}{\|\bar{c} + \Delta\|_{A^{-1}}^3}.$$

The following lemma simplifies $\nabla \ell_{\text{SPO+}}(\Delta)$.

**Lemma C.6.** *Suppose* $\mathbb{P} \in \mathcal{P}_{\text{rot symm}}$. *Then there exists a unique function* $\zeta(\cdot) : [0, \infty] \to [0, \infty]$ *such that for all* $\Delta \in \mathbb{R}^d$, *it holds that*

$$\mathbb{E}_c\left[\frac{c + \Delta}{\|c + \Delta\|_{A^{-1}}}\right] = \zeta(\|\bar{c} + \Delta\|_{A^{-1}})(\bar{c} + \Delta).$$

*Also,* $\alpha \cdot \zeta(\alpha)$ *is a non-decreasing function.*

*Proof.* Let $h(\Delta)$ denote $\mathbb{E}_c[\frac{c+\Delta}{\|c+\Delta\|}]$. First we show that $h(\Delta)$ has the same direction as $\bar{c} + \Delta$. Let $\phi_\Delta(\cdot)$ denote the affine transform $\phi_\Delta(\cdot) : \xi \to \frac{2(\bar{c}+\Delta)^T A^{-1}\xi}{\|\bar{c}+\Delta\|_{A^{-1}}^2}(\bar{c}+\Delta) - \xi$. We have $\phi_\Delta(\phi_\Delta(\xi)) = \xi$ and $\|\xi\|_{A^{-1}} = \|\phi_\Delta(\xi)\|_{A^{-1}}$ for all $\xi \in \mathbb{R}^d$. It leads to $p(\xi) = p(\phi_\Delta(\xi))$ and hence

$$h(\Delta) = \frac{1}{2}\mathbb{E}_c\left[\frac{c+\Delta}{\|c+\Delta\|_{A^{-1}}} + \frac{\phi_\Delta(c+\Delta)}{\|\phi_\Delta(c+\Delta)\|_{A^{-1}}}\right] = \frac{1}{2}\mathbb{E}_c\left[\frac{(c+\Delta) + \phi_\Delta(c+\Delta)}{\|c+\Delta\|_{A^{-1}}}\right]$$

$$= \mathbb{E}_c\left[\frac{(\bar{c}+\Delta)^T A^{-1}(c+\Delta)}{\|c+\Delta\|_{A^{-1}} \cdot \|\bar{c}+\Delta\|_{A^{-1}}^2}\right](\bar{c}+\Delta).$$

Now we let

$$\hat{\zeta}(\bar{c}+\Delta) = \mathbb{E}_c\left[\frac{(\bar{c}+\Delta)^T A^{-1}(c+\Delta)}{\|c+\Delta\|_{A^{-1}} \cdot \|\bar{c}+\Delta\|_{A^{-1}}^2}\right],$$

and we want to show that $\hat{\zeta}(\bar{c}+\Delta) = \hat{\zeta}(\bar{c}+\Delta')$ if $\|\bar{c}+\Delta\|_{A^{-1}} = \|\bar{c}+\Delta'\|_{A^{-1}}$. Since $\|\bar{c}+\Delta\|_{A^{-1}} = \|\bar{c}+\Delta'\|_{A^{-1}}$, there exists a matrix $R \in \mathbb{R}^{d\times d}$ such that $A^{-1/2}(\bar{c}+\Delta') = RA^{-1/2}(\bar{c}+\Delta)$ and $RR^T = R^T R = I$. Let $c'$ be a random variable depending on $c$ where $c' = A^{1/2}RA^{-1/2}(c-\bar{c}) + \bar{c}$. It holds that $A^{-1/2}(c'-\bar{c}) = RA^{-1/2}(c-\bar{c})$, which implies that $\|c'-\bar{c}\|_{A^{-1}} = \|c-\bar{c}\|_{A^{-1}}$ and therefore $p(c-\bar{c}) = p(c'-\bar{c})$. Also, we have $A^{-1/2}(c'+\Delta') = RA^{-1/2}(c+\Delta)$, which implies that $\|c'+\Delta'\|_{A^{-1}} = \|c+\Delta\|_{A^{-1}}$ and therefore

$$\frac{(\bar{c}+\Delta')^T A^{-1}(c'+\Delta')}{\|c'+\Delta'\|_{A^{-1}}} = \frac{(\bar{c}+\Delta)^T A^{-1/2}R^T RA^{-1/2}(c+\Delta)}{\|c+\Delta\|_{A^{-1}}} = \frac{(\bar{c}+\Delta)^T A^{-1}(c+\Delta)}{\|c+\Delta\|_{A^{-1}}}.$$

Moreover, since $\det(A^{1/2}RA^{-1/2}) = 1$, it holds that

$$\mathbb{E}_c\left[\frac{(\bar{c}+\Delta')^T A^{-1}(c+\Delta')}{\|c+\Delta'\|_{A^{-1}}}\right] = \mathbb{E}_c\left[\frac{(\bar{c}+\Delta')^T A^{-1}(c'+\Delta')}{\|c'+\Delta'\|_{A^{-1}}}\right] = \mathbb{E}_c\left[\frac{(\bar{c}+\Delta')^T A^{-1}(c+\Delta')}{\|c+\Delta'\|_{A^{-1}}}\right].$$

Therefore,

$$\hat{\zeta}(\bar{c}+\Delta) = \frac{1}{\|\bar{c}+\Delta\|_{A^{-1}}^2} \cdot \mathbb{E}_c\left[\frac{(\bar{c}+\Delta')^T A^{-1}(c+\Delta')}{\|c+\Delta'\|_{A^{-1}}}\right]$$

$$= \frac{1}{\|\bar{c}+\Delta'\|_{A^{-1}}^2} \cdot \mathbb{E}_c\left[\frac{(\bar{c}+\Delta')^T A^{-1}(c+\Delta')}{\|c+\Delta'\|_{A^{-1}}}\right] = \hat{\zeta}(\bar{c}+\Delta').$$

Therefore, we know that $\zeta(\cdot) : \mathbb{R} \to \mathbb{R}$ is a well-defined function based on the above property of $\hat{\zeta}(\cdot)$. Now we are going to prove that $\alpha \cdot \zeta(\alpha)$ is a non-decreasing function. Pick arbitrary $\alpha_1' > \alpha_2' > 0$, we have $\zeta(\alpha_1') = \hat{\zeta}(\alpha_1 \cdot \bar{c})$ and $\zeta(\alpha_2') = \hat{\zeta}(\alpha_2 \cdot \bar{c})$, where $\alpha_i = \alpha_i'/\|\bar{c}\|_{A^{-1}}$ for $i = 1, 2$. Therefore,

$$\alpha_1' \cdot \zeta(\alpha_1') \geq \alpha_2' \cdot \zeta(\alpha_2') \Leftrightarrow \alpha_1 \cdot \hat{\zeta}(\alpha_1 \cdot \bar{c}) \geq \alpha_2 \cdot \hat{\zeta}(\alpha_2 \cdot \bar{c})$$

$$\Leftrightarrow \alpha_1 \mathbb{E}_c\left[\frac{(\alpha_1 \cdot \bar{c})^T A^{-1}((c-\bar{c}) + \alpha_1 \cdot \bar{c})}{\|(c-\bar{c}) + \alpha_1 \cdot \bar{c}\|_{A^{-1}} \cdot \|\alpha_1 \cdot \bar{c}\|_{A^{-1}}^2}\right] \geq \alpha_2 \mathbb{E}_c\left[\frac{(\alpha_2 \cdot \bar{c})^T A^{-1}((c-\bar{c}) + \alpha_2 \cdot \bar{c})}{\|(c-\bar{c}) + \alpha_2 \cdot \bar{c}\|_{A^{-1}} \cdot \|\alpha_2 \cdot \bar{c}\|_{A^{-1}}^2}\right]$$

$$\Leftrightarrow \mathbb{E}_c\left[\frac{\bar{c}^T A^{-1}((c-\bar{c}) + \alpha_1 \cdot \bar{c})}{\|(c-\bar{c}) + \alpha_1 \cdot \bar{c}\|_{A^{-1}}}\right] \geq \mathbb{E}_c\left[\frac{\bar{c}^T A^{-1}((c-\bar{c}) + \alpha_2 \cdot \bar{c})}{\|(c-\bar{c}) + \alpha_2 \cdot \bar{c}\|_{A^{-1}}}\right].$$

It is sufficient to show that

$$\frac{\bar{c}^T A^{-1}(\zeta + \alpha_1 \cdot \bar{c})}{\|\zeta + \alpha_1 \cdot \bar{c}\|_{A^{-1}}} \geq \frac{\bar{c}^T A^{-1}(\zeta + \alpha_2 \cdot \bar{c})}{\|\zeta + \alpha_2 \cdot \bar{c}\|_{A^{-1}}}, \tag{13}$$

for all $\zeta \in \mathbb{R}^d$ when $\alpha_1 > \alpha_2 > 0$. We divide the proof into three cases. When $\bar{c}^T A^{-1}(\zeta + \alpha_1 \cdot \bar{c}) > \bar{c}^T A^{-1}(\zeta + \alpha_2 \cdot \bar{c}) \geq 0$, (13) is equivalent to

$$\left(\bar{c}^T A^{-1}(\zeta + \alpha_1 \cdot \bar{c})\right)^2 \cdot \|\zeta + \alpha_2 \cdot \bar{c}\|_{A^{-1}}^2 \geq \left(\bar{c}^T A^{-1}(\zeta + \alpha_2 \cdot \bar{c})\right)^2 \cdot \|\zeta + \alpha_1 \cdot \bar{c}\|_{A^{-1}}^2$$

$$\Leftrightarrow (\alpha_1 - \alpha_2)\left(\bar{c}^T A^{-1}(\zeta + \alpha_1 \cdot \bar{c}) + \bar{c}^T A^{-1}(\zeta + \alpha_2 \cdot \bar{c})\right)\left(\bar{c}^T A^{-1}\bar{c} \cdot \zeta^T A^{-1}\zeta - (\bar{c}^T A^{-1}\zeta)^2\right) \geq 0.$$

When $\bar{c}^T(\zeta + \alpha_1 \cdot \bar{c}) \geq 0 \geq \bar{c}^T(\zeta + \alpha_2 \cdot \bar{c})$, we know that left hand side of (13) is non-negative and right hand side is non-positive. When $0 > \bar{c}^T(\zeta + \alpha_1 \cdot \bar{c}) \geq \bar{c}^T(\zeta + \alpha_2 \cdot \bar{c})$, (13) is equivalent to

$$\left(\bar{c}^T A^{-1}(\zeta + \alpha_1 \cdot \bar{c})\right)^2 \cdot \|\zeta + \alpha_2 \cdot \bar{c}\|_{A^{-1}}^2 \leq \left(\bar{c}^T A^{-1}(\zeta + \alpha_2 \cdot \bar{c})\right)^2 \cdot \|\zeta + \alpha_1 \cdot \bar{c}\|_{A^{-1}}^2$$

$$\Leftrightarrow (\alpha_1 - \alpha_2)\left(\bar{c}^T A^{-1}(\zeta + \alpha_1 \cdot \bar{c}) + \bar{c}^T A^{-1}(\zeta + \alpha_2 \cdot \bar{c})\right)\left(\bar{c}^T A^{-1}\bar{c} \cdot \zeta^T A^{-1}\zeta - (\bar{c}^T A^{-1}\zeta)^2\right) \leq 0.$$

$\square$

Following the results in Lemma C.6 we have

$$\mathbb{E}_c \left[ \frac{c}{\|c\|_{A^{-1}}} \right] = \zeta(\|\bar{c}\|_{A^{-1}})\bar{c}, \quad \mathbb{E}_c \left[ \frac{c + 2\Delta}{\|c + 2\Delta\|_{A^{-1}}} \right] = \zeta(\|\bar{c} + 2\Delta\|_{A^{-1}})(\bar{c} + 2\Delta).$$

Hence, (12) is equivalent to

$$\zeta(\|\bar{c} + 2\Delta\|_{A^{-1}})(\bar{c} + 2\Delta) - \zeta(\|\bar{c}\|_{A^{-1}})\bar{c} = \alpha \cdot \frac{\bar{c}^T A^{-1}(\bar{c} + \Delta) \cdot \Delta - \Delta^T A^{-1}(\bar{c} + \Delta) \cdot \bar{c}}{\|\bar{c} + \Delta\|_{A^{-1}}^3}.$$

Since $\bar{c}$ and $\Delta$ are linearly independent, (12) is further equivalent to

$$\frac{2\zeta(\|\bar{c} + 2\Delta\|_{A^{-1}})}{\bar{c}^T A^{-1}(\bar{c} + \Delta)} = \frac{\alpha}{\|\bar{c} + 2\Delta\|_{A^{-1}}^3} = \frac{\zeta(\|\bar{c} + 2\Delta\|_{A^{-1}}) - \zeta(\|\bar{c}\|_{A^{-1}})}{-\Delta^T A^{-1}(\bar{c} + \Delta)},$$

which is also equivalent to

$$(\bar{c} + 2\Delta)^T A^{-1}(\bar{c} + \Delta) \cdot \zeta(\|\bar{c} + 2\Delta\|_{A^{-1}}) = \bar{c}^T A^{-1}(\bar{c} + \Delta) \cdot \zeta(\|\bar{c}\|_{A^{-1}}). \tag{14}$$

**Lemma C.7.** *Suppose $\mathbb{P} \in \mathcal{P}_{\text{rot symm}}$ and $\hat{\Delta}$ is a solution to (12), then it holds that*

$$\|\bar{c} + 2\hat{\Delta}\|_{A^{-1}} = \|\bar{c}\|_{A^{-1}},$$

*and*

$$(\bar{c} + 2\hat{\Delta})^T A^{-1}(\bar{c} + \hat{\Delta}) = \bar{c}^T A^{-1}(\bar{c} + \hat{\Delta}).$$

*Proof.* Suppose $\|\bar{c} + 2\hat{\Delta}\|_{A^{-1}} \neq \|\bar{c}\|_{A^{-1}}$. Without loss of generality we assume $\|\bar{c} + 2\hat{\Delta}\|_{A^{-1}} > \|\bar{c}\|_{A^{-1}}$. Following results in Lemma C.6 we know that

$$\|\bar{c} + 2\Delta\|_{A^{-1}} \cdot \zeta(\|\bar{c} + 2\Delta\|_{A^{-1}}) \geq \|\bar{c}\|_{A^{-1}} \cdot \zeta(\|\bar{c}\|_{A^{-1}}).$$

Also, it holds that

$$\hat{\Delta}^T A^{-1}(\bar{c} + \hat{\Delta}) = \frac{1}{4} \left( \|\bar{c} + 2\hat{\Delta}\|_{A^{-1}}^2 - \|\bar{c}\|_{A^{-1}}^2 \right) > 0.$$

Since $(\bar{c} + 2\hat{\Delta})^T A^{-1}(\bar{c} + \hat{\Delta}) = (\bar{c} + \hat{\Delta})^T A^{-1}(\bar{c} + \hat{\Delta}) + \hat{\Delta}^T A^{-1}(\bar{c} + \hat{\Delta}) > 0$, it holds that

$$\frac{(\bar{c} + 2\hat{\Delta})^T A^{-1}(\bar{c} + \hat{\Delta})}{\|\bar{c} + 2\hat{\Delta}\|_{A^{-1}}} > \frac{\bar{c}^T A^{-1}(\bar{c} + \hat{\Delta})}{\|\bar{c}\|_{A^{-1}}}$$

$$\Leftrightarrow (\bar{c} + 2\hat{\Delta})^T A^{-1}(\bar{c} + \hat{\Delta}) \cdot \|\bar{c}\|_{A^{-1}} > \bar{c}^T A^{-1}(\bar{c} + \hat{\Delta}) \cdot \|\bar{c} + 2\hat{\Delta}\|_{A^{-1}}$$

$$\Leftarrow \left( (\bar{c} + 2\hat{\Delta})^T A^{-1}(\bar{c} + \hat{\Delta}) \right)^2 \cdot \|\bar{c}\|_{A^{-1}}^2 > \left( \bar{c}^T A^{-1}(\bar{c} + \hat{\Delta}) \right)^2 \cdot \|\bar{c} + 2\hat{\Delta}\|_{A^{-1}}^2$$

$$\Leftrightarrow \left( \hat{\Delta}^T A^{-1}(\bar{c} + \hat{\Delta}) \right) \cdot \left( \|\bar{c} + \hat{\Delta}\|_{A^{-1}}^2 \cdot \|\hat{\Delta}\|_{A^{-1}}^2 - (\hat{\Delta}^T A^{-1}(\bar{c} + \hat{\Delta}))^2 \right) > 0.$$

Therefore, we have

$$(\bar{c} + 2\Delta)^T A^{-1}(\bar{c} + \Delta) \cdot \zeta(\|\bar{c} + 2\Delta\|_{A^{-1}}) > \bar{c}^T A^{-1}(\bar{c} + \Delta) \cdot \zeta(\|\bar{c}\|_{A^{-1}}),$$

which contradicts with (14). Therefore, we have $\|\bar{c} + 2\hat{\Delta}\|_{A^{-1}} = \|\bar{c}\|_{A^{-1}}$ and hence $(\bar{c} + 2\hat{\Delta})^T A^{-1}(\bar{c} + \hat{\Delta}) = \bar{c}^T A^{-1}(\bar{c} + \hat{\Delta})$. $\qquad\square$

Based on the above property, we provide a lower bound of calibration function.

**Theorem C.3.** *Suppose Assumption C.1 holds and $\mathbb{P} \in \mathcal{P}_{\text{rot symm}}$, then the calibration function $\delta(\cdot)$ satisfies*

$$\delta(\epsilon) \geq \mathbb{E}_c \left[ \frac{\min\{\|\bar{c}\|_{A^{-1}}, \|c - \bar{c}\|_{A^{-1}}\}}{\|c - \bar{c}\|_{A^{-1}}^2 + \|\bar{c}\|_{A^{-1}}\|c - \bar{c}\|_{A^{-1}}} \right] \cdot \frac{\mu^{9/2}\|\bar{c}\|_{A^{-1}}}{2L^{9/2}} \cdot \epsilon,$$

*for all $\epsilon > 0$.*

*Proof.* First we know that $\delta(\epsilon) \geq \delta'(\epsilon)$. Also, Lemma C.7 shows that for optimal $\Delta$, it holds that $\|\bar{c}\|_{A^{-1}} = \|\bar{c} + 2\Delta\|_{A^{-1}}$. By the definition of $\underline{R}_{\text{SPO+}}$, we have

$$\underline{R}_{\text{SPO+}}(\Delta) = \mathbb{E}_c[\ell_{\text{SPO+}}(c, \Delta)] = \mathbb{E}_c\left[\|c + 2\Delta\|_{A^{-1}} - \frac{c^T A^{-1}(c + 2\Delta)}{\|c\|_{A^{-1}}}\right]$$

$$= \mathbb{E}_c\left[\|c + 2\Delta\|_{A^{-1}}\right] - \mathbb{E}_c\left[\|c\|_{A^{-1}}\right] - \mathbb{E}_c\left[\frac{2c^T A^{-1}\Delta}{\|c\|_{A^{-1}}}\right].$$

Since $\|\bar{c} + 2\Delta\|_{A^{-1}} = \|\bar{c}\|_{A^{-1}}$, Proposition C.1 shows that $\mathbb{E}_c[\|c + 2\Delta\|_{A^{-1}}] = \mathbb{E}_c[\|c\|_{A^{-1}}]$. Therefore, it holds that

$$\underline{R}_{\text{SPO+}}(\Delta) = -\mathbb{E}_c\left[\frac{2c^T A^{-1}\Delta}{\|c\|_{A^{-1}}}\right] = -\mathbb{E}_c\left[\frac{(c + \phi_0(c))^T A^{-1}\Delta}{\|c\|_{A^{-1}}}\right]$$

$$= \mathbb{E}_c\left[\frac{\bar{c}^T A^{-1} c}{\|\bar{c}\|_{A^{-1}}^2} \cdot \frac{\bar{c}^T A^{-1}\Delta}{\|c\|_{A^{-1}}}\right] = \mathbb{E}_c\left[\frac{\bar{c}^T A^{-1} c}{\|c\|_{A^{-1}}}\right] \cdot \frac{-\bar{c}^T A^{-1}\Delta}{\|\bar{c}\|_{A^{-1}}^2}$$

$$= \mathbb{E}_c\left[\frac{\bar{c}^T A^{-1} c}{\|c\|_{A^{-1}}}\right] \cdot \frac{\Delta^T A^{-1}\Delta}{\|\bar{c}\|_{A^{-1}}^2},$$

where the last inequality holds since $(\bar{c} + \Delta)^T A^{-1}\Delta = 0$. Based on the result in Proposition C.2, we have

$$\mathbb{E}_c\left[\frac{\bar{c}^T A^{-1} c}{\|c\|_{A^{-1}}}\right] \geq \mathbb{E}_c \frac{\|c\|_{A^{-1}}^2 \min\{\|c\|_{A^{-1}}, \|c - \bar{c}\|_{A^{-1}}\}}{\|c - \bar{c}\|_{A^{-1}}^2 + \|\bar{c}\|_{A^{-1}}\|c - \bar{c}\|_{A^{-1}}}.$$

Also, let $\epsilon' = C_2(\mu, L, r) \cdot \epsilon$. In the constraint we have

$$\|\bar{c}\|_{A^{-1}} - \frac{\bar{c}^T A^{-1}(\bar{c} + \Delta)}{\|\bar{c} + \Delta\|_{A^{-1}}} \geq \epsilon',$$

and hence $\|\bar{c}\|_{A^{-1}} - \|\bar{c} + \Delta\|_{A^{-1}} \geq \epsilon'$. Since $\|\bar{c}\|_{A^{-1}} \geq \epsilon'$, it holds that $(\|\bar{c}\|_{A^{-1}} - \epsilon')^2 \geq \|\bar{c} + \Delta\|_{A^{-1}}^2$. This implies that $\Delta^T A^{-1}\Delta \geq 2\|\bar{c}\|_{A^{-1}}\epsilon' - \epsilon'^2 \geq \|\bar{c}\|_{A^{-1}}\epsilon' = \|\bar{c}\|_{A^{-1}}C_2(\mu, L, r)\epsilon$. Therefore, we conclude that

$$\delta(\epsilon) \geq \mathbb{E}_c\left[\frac{\min\{\|\bar{c}\|_{A^{-1}}, \|c - \bar{c}\|_{A^{-1}}\}}{\|c - \bar{c}\|_{A^{-1}}^2 + \|\bar{c}\|_{A^{-1}}\|c - \bar{c}\|_{A^{-1}}}\right] \cdot \frac{\mu^{9/2}\|\bar{c}\|_{A^{-1}}}{2L^{9/2}} \cdot \epsilon.$$

$\square$

We are now ready to complete the proof of Theorem C.1.

*Proof of Theorem C.1.* From Theorem C.3, we know that

$$\delta(\epsilon; x, \mathbb{P}) \geq \mathbb{E}_{c|x}\left[\frac{\min\{\|\bar{c}\|_{A^{-1}}, \|c - \bar{c}\|_{A^{-1}}\} \cdot \|\bar{c}\|_{A^{-1}}}{\|c - \bar{c}\|_{A^{-1}}^2 + \|\bar{c}\|_{A^{-1}}\|c - \bar{c}\|_{A^{-1}}}\right] \cdot \frac{\mu^{9/2}\epsilon}{2L^{9/2}}.$$

Also, by $\frac{\min\{c_1, c_2\} \cdot c_1}{c_2^2 + c_1 c_2} \geq \frac{c_1^2}{2(c_1^2 + c_2^2)}$ for all $c_1, c_2 \neq 0$, we have

$$\delta(\epsilon; x, \mathbb{P}) \geq \mathbb{E}_{c|x}\left[\frac{\|\bar{c}\|_{A^{-1}}^2}{2(\|\bar{c}\|_{A^{-1}}^2 + \|c - \bar{c}\|_{A^{-1}}^2)}\right] \cdot \frac{\mu^{9/2}\epsilon}{2L^{9/2}}. \tag{15}$$

Moreover, for all $\mathbb{P} \in \mathcal{P}_{\alpha,\beta}$, it holds that

$$\mathbb{E}_{c|x}\left[\frac{\|\bar{c}\|_{A^{-1}}^2}{\|\bar{c}\|_{A^{-1}}^2 + \|c - \bar{c}\|_{A^{-1}}^2}\right] \geq \mathbb{E}_{c|x}\left[\frac{\|\bar{c}\|_{A^{-1}}^2}{\|\bar{c}\|_{A^{-1}}^2 + \|c - \bar{c}\|_{A^{-1}}^2}\,\middle|\, \|c - \bar{c}\|_{A^{-1}} \leq \beta \cdot \|\bar{c}\|_{A^{-1}}\right]$$

$$\cdot \mathbb{P}_{c|x}(\|c - \bar{c}\|_{A^{-1}} \leq \beta \cdot \|\bar{c}\|_{A^{-1}})$$

$$\geq \frac{\alpha}{1 + \beta^2},$$

and for all $\mathbb{P} \in \mathcal{P}_\beta$, it holds that

$$\mathbb{E}_{c|x}\left[\frac{\|\bar{c}\|_{A^{-1}}^2}{\|\bar{c}\|_{A^{-1}}^2 + \|c - \bar{c}\|_{A^{-1}}^2}\right] \geq \frac{\|\bar{c}\|_{A^{-1}}^2}{\|\bar{c}\|_{A^{-1}}^2 + \mathbb{E}_{c|x}[\|c - \bar{c}\|_{A^{-1}}^2]} \geq \frac{1}{1 + \beta^2}.$$

By applying the above two inequalities to (15) we complete the proof. $\square$

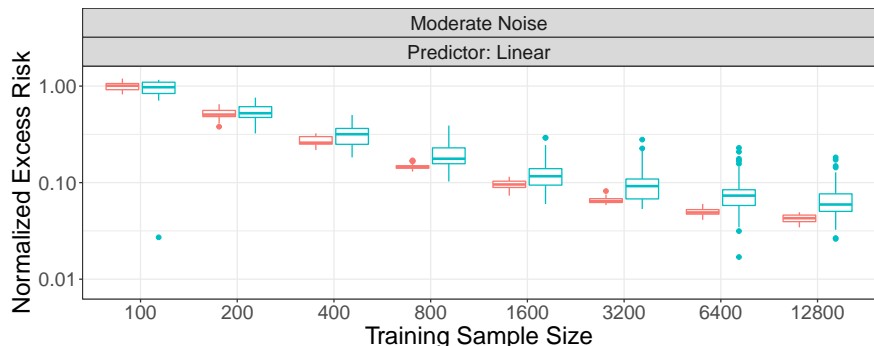

Figure 3: Normalized test set excess risk for the SPO+ methods on instances with polyhedron and level-set feasible regions. For each value of the sample size in the above plots we run $50$ independent trials.

## D   Experimental details

For both problems, we ran each instance on one core of Intel Xeon Skylake 6230 @ 2.1 GHz.

### D.1   Excess risk comparison

In Figure 3, we provide the empirical excess risk comparison of the cases with polyhedral and level-set feasible regions. The case with polyhedral feasible region are the cost-sensitive multi-class classification instances with simplex feasible region, and the case with level-set feasible region are the entropy constrained portfolio optimization problems. The main metric we use in Figure 3 is the *normalized excess risk*, which for each case, is defined as the excess risk over the averaged excess risk with sample size $n = 100$. For each type of feasible region, the excess risk is calculated by the difference between the SPO risk of the predictions given by the trained model and the true model. Also, we set polynomial degree equals to one with moderate noises, which means the true model is in the hypothesis class. The main purpose of this plot is not checking if the order of the calibration matches the theoretical results, as these are only worst case guarantees, but qualitatively comparing the convergence of excess risk with different types of feasible regions.

### D.2   Additional plots on the cost-sensitive multi-class classification instances

In Figure 4, we provide a complete comparison of all the method on the cost-sensitive multi-class classification instances. We can observe a similar pattern as in Figure 1.

### D.3   Technical details

In Lemma D.1 we show that the optimization oracle $w^*(\cdot)$ is differentiable when the projection of the predicted cost vector $\hat{c}$ is not zero for the entropy constrained portfolio optimization example.

**Lemma D.1.** *Let $T = \{w \in \mathbb{R}^d : w > 0, \mathbf{1}^T w = 1\}$ denote the interior of the probability simplex. For any vector $c \in \mathbb{R}^d$, let $\tilde{c}$ denote the projection of $c$ onto $T$. Let $f(w) = \sum_{i=1}^d -w_i \log(w_i)$ denote the entropy function. For some scalar $r \in (f_{\min}, \varliminf_{w \to \partial T} f(w))$, let $S = \{w \in T : f(w) \leq r\}$. Let $w^*(c) = \arg\min_{w \in S} c^T w$. Then it holds that $w^*(c)$ is differentiable when $\tilde{c} \neq 0$ where $\tilde{c}$ is the projection of $c$ onto the subspace $\{w \in \mathbb{R}^d : \mathbf{1}^T w = 0\}$.*

*Proof.* Let $\mathrm{softmax}(\cdot) : \mathbb{R}^d \to \mathbb{R}^d$ denote the softmax function, namely

$$\mathrm{softmax}(c) = \left[ \frac{\exp(c_1)}{\sum_{i=1}^d \exp(c_i)}, \ldots, \frac{\exp(c_d)}{\sum_{i=1}^d \exp(c_i)} \right]^T.$$

## Normalized SPO Loss v.s. Polynomial Degree

Figure 4: Test set SPO loss for the SPO+, least squares, and absolute loss methods on cost-sensitive multi-class classification instances. For each value of the polynomial degree in the above plots we run $50$ independent trials.

Using KKT condition, we know that for any $c \in \mathbb{R}^d$ such that $\tilde{c} \neq 0$, there exists some scalar $u(c) \geq 0$ such that $c = -u(c) \cdot \nabla f(w^*(c))$, and therefore $w^*(c) = \text{softmax}(-\tilde{c}/u(c))$. Since the softmax function is differentiable and $\tilde{c}$ is differentiable with respect to $c$, we only need to show that the function $u(c)$ is also differentiable with respect to $c$. Indeed, when $\tilde{c} \neq 0$, we have $f(w^*(c)) = r$, which is equivalent to $f(\text{softmax}(-\tilde{c}/u(c))) = r$. Let $\phi(c, u) = f(\text{softmax}(-\tilde{c}/u))$. Since $\phi(c, u)$ is a decreasing function for $u > 0$, by inverse function theorem we have $\frac{du}{dc} = -\frac{\partial \phi/\partial c}{\partial \phi/\partial u}$, and hence $u(c)$ is also differentiable with respect to $c$. $\qquad\square$

In the cost-sensitive multi-class classification problem, we consider the SPO+ method using a log barrier approximation to the unit simplex. For the choice of the threshold $r$, according to Assumption C.1 we will need $r > f_{\min}$ and $r < \underline{\lim}_{w \to \partial T} f(w)$. In this log barrier scenario, we have $f_{\min} = d \log d$ and $\underline{\lim}_{w \to \partial T} f(w) = \infty$. Therefore, we pick the threshold $r = 2d \log d$. Of course, one may consider a more careful tuning of this hyper-parameter. Nevertheless, even with our simplistic approach for setting it we observe benefits of the SPO+ loss that uses a log barrier approximation to the unit simplex.

### D.4 Data generation processes

In the next two paragraphs we discuss the detailed data generation process of each problem.

**Portfolio allocation problems.** Let us describe the process used for generating the synthetic data sets for portfolio allocation instances. In this experiment, we set the number of assets $d = 50$ and the dimension of feature vector $p = 5$. We first generate a weight matrix $B \in \mathbb{R}^{d \times p}$, whereby each entry of $B$ is a Bernoulli random variable with the probability $\mathbb{P}(B_{ij} = 1) = \frac{1}{2}$. We then generate the training data set $\{(x_i, c_i)\}_{i=1}^n$ and the test data set $\{(\tilde{x}_i, \tilde{c}_i)\}_{i=1}^m$ independently according to the following procedure.

1. First we generate the feature vector $x \in \mathbb{R}^p$ from the standard multivariate normal distribution, namely $x \sim \mathcal{N}(0, I_p)$.

2. Then we generate the true cost vector $c \in \mathbb{R}^d$ according to $c_j = \left[ 1 + \left( 1 + \frac{b_j^T x}{\sqrt{p}} \right)^{\deg} \right] \epsilon_j$ for

   $j = 1, \ldots, d$, where $b_j$ is the $j$-th row of matrix $B$. Here deg is the fixed degree parameter and $\epsilon_j$, the multiplicative noise term, is a random variable which independently generated

from the uniform distribution $[1 - \bar{\epsilon}, 1 + \bar{\epsilon}]$ for a fixed noise half width $\bar{\epsilon} \geq 0$. In particular, $\bar{\epsilon}$ is set to 0 for "no noise" instances and 0.5 for "moderate noise" instances.

**Cost-sensitive multi-class classification problems.** Let us describe the process used for generating the synthetic data sets for cost-sensitive multi-class classification instances. In this experiment, we set the number of class $d = 10$ and the dimension of feature vector $p = 5$. We first generate a weight vector $b \in \mathbb{R}^p$, whereby each entry of $b$ is a Bernoulli random variable with the probability $\mathbb{P}(b_j = 1) = \frac{1}{2}$. We then generate the training data set $\{(x_i, c_i)\}_{i=1}^n$ and the test data set $\{(\tilde{x}_i, \tilde{c}_i)\}_{i=1}^m$ independently according to the following procedure.

1. First we generate the feature vector $x \in \mathbb{R}^p$ from the standard multivariate normal distribution, namely $x \sim \mathcal{N}(0, I_p)$.

2. Then we generate the score $s \in (0, 1)$ according to $s = \sigma \left( (b^T x)^{\mathrm{deg}} \cdot \mathrm{sign}(b^T x) \cdot \epsilon \right)$, where $\sigma(\cdot)$ is the logistic function. Here $\epsilon$, the multiplicative noise term, is a random variable which independently generated from the uniform distribution $[1 - \bar{\epsilon}, 1 + \bar{\epsilon}]$ for a fixed noise half width $\bar{\epsilon} \geq 0$. In particular, $\bar{\epsilon}$ is set to 0 for "no noise" instances and 0.5 for "moderate noise" instances.

3. Finally we generate the true class label $\mathrm{lab} = \lceil 10s \rceil \in \{1, \ldots, 10\}$ and the true cost vector $c = (c_1, \ldots, c_{10})$ is given by $c_j = |j - \mathrm{lab}|$ for $j = 1, \ldots, 10$.