# OpenReview forum: "Risk Bounds and Calibration for a Smart Predict-then-Optimize Method"
_NeurIPS.cc/2021/Conference — NeurIPS 2021 Poster_

### Official Review · Reviewer_FdGF · 2021-07-16

**Rating:** 6
**Confidence:** 4

**Summary:**

This paper derives uniform calibration bounds for the SPO+ loss relative to the SPO loss, when the feasible region is polyhedral or a level set of strongly convex functions. This enables us to translate excessive risk bound in terms of SPO+ loss, ie, the convex surrogate loss,  into bounds for the SPO loss. Notably, the authors show that the class of symmetric distributions considered in Elmachtoub and Grigas is not restrictive enough to derive calibration bounds, and instead stronger restrictions must be incorporated.

**Limitations And Societal Impact:**

I don't think that this work will have negative societal impact.

**Main Review:**

This paper looks very solid. It studies a very important question about the SPO method that attracts a lot of attention recently, ie, whether small risks for the computationally tractable surrogate loss imply small risks for the loss of interest.

Although that this paper can derive some calibration results is really nice, I am not sure how relevant the results are for explaining the use of SPO+ loss.
	1. In particular, the calibration results in this paper are all based on some restrictions on the true data distributions. But this paper has very little discussions about these assumptions. For example, how strong are these assumptions? What do some parameters (eg, alpha, beta..) mean? How do they influence the final bounds? What is the intuition? What if the assumptions are violated?
	2. The Experiments in this paper are very disconnected from the theory. I don't see how the experiments even try to support any of the theoretical findings. It seems that the current experiments largely just shows that SPO works better, especially when models are misspecified, which is similar to the original SPO paper. But what's new?  Actually, experiments might be a very good way to assess the those distributional assumptions.
-------------------
Considering the authors' response, I decide to change my score from 5 to 6.

**Time Spent Reviewing:**

2

---

> ### Author Response · Authors · 2021-08-10
> **Response to Reviewer FdGF**
>
> Thanks for the insightful review. We really appreciate the comments and suggestions, which we believe will help to improve the quality of the paper.
>
> **Distributional Assumptions/Parameters:** We have included a few sentences discussing the assumptions throughout the paper, and we will expand upon this in the camera-ready version. Regarding the strength of the assumptions, many of the common distributions satisfy the assumptions, including the multi-variate normal, Laplacian, Cauchy, and uniform distribution on a bounded set (as discussed in the paper).
>     The main idea of the assumptions is that we want to avoid a situation where the density of the cost vector concentrates around some "badly behaved points". Where these "badly behaved points" are located depends on the structure of $S$; Example 1 in the paper highlights a situation that considers one such case where a limiting distribution of a mixture of two Gaussians leads to a zero calibration function.
>     We believe that our assumptions are sufficiently general to cover many interesting and important cases. On the other hand, they may be violated in which case calibration may or may not hold.
>     A very interesting open question for future research is to also characterize necessary conditions for calibration of SPO+. Successfully answering that question would provide additional guidance on when one should use SPO+ versus a more traditional prediction based loss. Nevertheless, our results show that SPO+ is effective in many reasonable scenarios.
>
> In addition, we will add more discussion about the intuition of the parameters and how they impact the risk bounds.
>     Briefly, in the LP case, $\alpha$ is a lower bound on the ratio of density of the distribution of the cost vector relative to a "reference" standard normal distribution. When $\alpha$ is larger, the distribution is behaved more like a normal distribution and it leads to a better risk bound. $M$ is an upper bound on the standard deviation of the aforementioned reference normal distribution, and the bound naturally becomes worse as $M$ increases.
>     $\beta$ measures how the conditional mean deviates from zero relative to the standard deviation of the reference normal distribution. If this distance is larger then the predictions are larger on average and the risk bound becomes worse.
>     $d_S$ measures the near-degeneracy of the polyhedron ($d_S = 0$ is degenerate) and the bound becomes meaningless as $d_S \to 0$.
>     In the strongly convex set case, $\alpha$ is the probability that the cost vector is "relatively close'' to the conditional mean. When $\alpha$ is larger, the cost vector is more likely to be close to the conditional mean and the risk bound will be better.
>     $\beta$ controls the concentration of the distribution of cost vector around the conditional mean. The more concentrated the distribution is, the better the risk bound is.
>
> **Experiments:** There are a few novelties of our experiments that we would like to highlight. First, to the best of our knowledge, we are the first to test the SPO+ loss on a strongly convex set and the first to introduce the idea using the SPO+ loss defined by a (strongly convex) barrier approximation of a polyhedral set.
>
> Overall, our philosophy of the numerical experiments is that we want to design experiments that can:
>     (i) relate the theoretical findings to practice;
>     (ii) practically examine which method/loss to use to achieve small SPO loss under different scenarios.
>     We not only want to verify the theoretical results but also care about the practical consequences of them. For instance, an interesting question is: does the fact that the risk bound is faster in the strongly convex set case have any practical consequences?
>     In Figure 2, we designed the experiment to see whether we can benefit from the theoretical results in the strongly convex set case and we observe that using the barrier function is beneficial when the training set size is small. We believe that the results in Figure 2 highlight that -- as suggested by the theory -- using a strongly convex set in place of a polyhedral one *does* lead to improved results. (Note that these improved results are measured with respect to the SPO loss of the original polyhedral set, thus there is a trade-off depending on the sample size as we point out on page 9.)
>
> We do appreciate the suggestion that we could conduct additional experiments to assess the distributional assumptions. We do note that there is no guarantee that the distributions we consider in the experiments satisfy the theoretical assumptions in the paper. In fact, the only case that is guaranteed to satisfy the assumptions is the "no noise" case in Figure 1. Thus we believe that our results show some robustness to the theoretical distributional assumptions. Of course, as mentioned previously, finding necessary conditions for calibration is a very interesting open question and additional numerical experiments may provide guidance in this direction.

---

### Official Review · Reviewer_7rEw · 2021-07-16

**Rating:** 7
**Confidence:** 2

**Summary:**

The paper develops the risk bounds and uniform calibration results for the surrogate SPO+ loss relative to the SPO loss in the predict-then-optimize framework under the assumption of a linear loss function. The results are first obtained for a polyhedron feasible region of the optimization problem and further improved when the feasible region is a level set of a strongly convex function. Then, they are used to establish a sample complexity (with respect to the SPO loss) of the algorithm that minimizes the empirical SPO+ risk. The empirical performance of the SPO+ surrogate is demonstrated in portfolio optimization and cost-sensitive multi-class classification problems.

**Main Review:**

I believe that the paper presents an important extension of the existing Fisher consistency property of SPO+ w.r.t SPO with the risk bounds and uniform calibration results. This is because the Fisher consistency is not applicable when only finite datasets are available as explained in the paper. The assumption of linear loss function is also reasonable according to the presented literature of the predict-then-optimize framework. Overall, the paper is well-written and well-structured.

I only have minor questions related to the experiments and the fact that the conclusion section is missing. In figure 1, when the predictor is a neural network and the polynomial degree is 6 or 8, the performance of SPO+ is only on par with l1 loss and is outperformed significantly by SPO. This is different from the case when the predictor is a linear model. What could be the reason for this? I think that the assumption $g^*(x) = \mathbb{E}[c|x]$ (in Corollaries 3.1 and 4.1) is violated more severely for simple predictors (e.g., linear models), so what does this imply?


**Time Spent Reviewing:**

12

---

> ### Author Response · Authors · 2021-08-10
> **Response to Reviewer 7rEw**
>
> Thanks for the insightful review. We really appreciate the comments and suggestions, which we believe will help to improve the quality of the paper.
>     We removed the conclusion section for space reasons and will add it into the extra page of the camera-ready version.
>
> Regarding the experiments, we also find the stark difference between SPO and SPO+, in the neural net case with large polynomial degree, to be very interesting. One plausible explanation for this behavior is as follows. Intuitively when the predictor is a linear model, most of the error comes from approximation error. Furthermore, since the hypothesis class is relatively small, it is reasonable to think that the linear models returned by SPO and SPO+ are close to each other, e.g., in Frobenius norm. This would have the consequence that the performances of the two loss functions (w.r.t. to the SPO loss) are both close to each other and essentially tied for the best, as we observe in the first row of Figure 1.
>
> On the other hand, for the neural nets case, we widen the hypothesis class significantly and it is less likely that the weights of the two models, one using SPO loss and the other SPO+, are close to each other. Moving to the neural nets case also has other consequences. First, the approximation error is closer to zero (as you point out) but the generalization/estimation error is larger than the linear case due to the richness of the hypothesis class. Another consequence of moving to the neural nets case is that the calibration results begin to "kick in". However, it is important to note that the calibration function only provides an upper bound and, due to the estimation error that is present, we need a sufficient amount of data to ensure that minimizing SPO+ also approximately minimizes SPO. Indeed, we do observe in the second row of Figure 1 that moving from 100 to 1000 samples slightly improves the performance of SPO+. We believe that with larger and larger data set sizes, SPO+ will eventually converge to the same level of performance as SPO. It would be interesting and worthwhile to experimentally try larger sample sizes in the future to verify these claims.

---

> > ### Comment · Reviewer_7rEw · 2021-08-19
> > **Response**
> >
> > Thank you for your detailed response which clears my doubt on the experiments.
> > It is indeed interesting to empirically validate your explanation regarding the performance of SPO+ vs. that of SPO with a large dataset in the revised paper.
> > I retain my recommendation for the paper.

---

### Official Review · Reviewer_cSZd · 2021-07-19

**Rating:** 7
**Confidence:** 4

**Summary:**

The paper studies the excess SPO risk of the empirical SPO+ risk minimizers. When the feasible region of the downstream planning task is a bounded polyhedron, the excess risk is on the order of O(n^{-1/4}). When the feasible region is the level set of a strongly convex and smooth function, the excess risk is on the order of O(n^{-1/2}). Empirically, the authors show that the empirical minimizers of SPO+ perform better than the ones for \ell_1, \ell_2 for more complex hypothesis class.

**Limitations And Societal Impact:**

The authors have adequately addressed the limitations of their work.

**Main Review:**

The paper is very clearly written with a detailed background on smart Predict-then-Optimize and the SPO+ surrogate loss. In general, the current manuscript could be improved in the following ways:

- The distributional assumptions for Theorem 3.1/Corollary 3.1: It is a bit unnatural to see that the assumption of the distribution requires P(c|x) to be heavy-tailed (in the sense that the tails are thicker than the ones for sub-Gaussians) to obtain the excess-risk guarantees. Intuitively, when P(c|x) is sub-Gaussian, the learning problem becomes much simpler and one would expect the excess risk of any proposed learning-planning scheme to be lower. Does this result suggest that when the data is nicely distributed (e.g., when P(c|x) is sub-Gaussian), one should use other losses instead of SPO+ loss or maybe just directly perform the traditional two-stage process: first learning then planning?
- For the experiment section, since the theoretical results are on excess risk---it would be nice/necessary to have an experiment that shows the excessive risk of empirical SPO+ minimizers against the number of samples and see whether the rate is on the order of O(n^{-1/4}) or O(n^{-1/2}).

*Update after author response* I have read the author response and I am satsfied with the answers. Hence, I maintain my score for acceptance.

**Time Spent Reviewing:**

2 hours

---

> ### Author Response · Authors · 2021-08-10
> **Response to Reviewer cSZd**
>
> Thanks for the insightful review. We really appreciate the comments and suggestions, which we believe will help to improve the quality of the paper.
>
> **The distributional assumptions for Theorem 3.1/Corollary 3.1:** The key idea of this assumption on the distribution is not about whether the distribution is thicker-tailed than sub-Gaussian or not (in fact Gaussian distributions satisfy the assumption). Rather, the main idea of the assumption is that we want to avoid a situation where the density of the cost vector concentrates around some "badly behaved points". Where these "badly behaved points" are located depends on the structure of $S$; Example 1 in the paper highlights a situation that considers one such case where a limiting distribution of a mixture of two Gaussians leads to a zero calibration function. We also would like to emphasize that our assumption provides generic sufficient conditions for calibration of the SPO+ loss. Thus, our results say something about when SPO+ is effective but do not characterize cases when it is not. A very interesting open question for future research is to also characterize necessary conditions. Successfully answering that question would provide additional guidance on when one should use SPO+ versus a more traditional prediction based loss. Finally, we think that in practice a combination of SPO+ with a traditional prediction loss (e.g., least squares) may offer the "best of both worlds" and be very effective.
>
>
> **Experiments:**
>     Since we generated the data from artificial distributions, it is easy to modify our experiments to show the excess risk. In the camera-ready version we will add these plots.
>     We chose not to include these results in the current version since:
>     (i) the theoretical results may (and often do) deviate from practice since the generalization bounds and calibration functions are only worst-case guarantees;
>     (ii) there may be faster rates when the distribution is well conditioned (one example is the ``low-noise'' condition in [Bartlett, Jordan and McAuliffe (2006)]).
>     Overall, our philosophy of the numerical experiments is that we want to design experiments that can:
>     (i) relate the theoretical findings to practice;
>     (ii) practically examine which method/loss to use to achieve small SPO loss under different scenarios.
>     We not only want to verify the theoretical results but also care about the practical consequences of them. For instance, an interesting question is: does the fact that the risk bound is faster in the strongly convex set case have any practical consequences?
>     In Figure 2, we designed the experiment to see whether we can benefit from the theoretical results in the strongly convex set case and we observe that using the barrier function is beneficial when the training set size is small.

---

### Decision · Program_Chairs · 2021-09-27

**Decision:**

Accept (Poster)

**Comment:**

This paper builds upon a prior work on Smart Predict-Then-Optimize (SPO) framework. Many practical problems can be described as 2 step procedures: 1. predict some quantity 2. solve optimization problem with predictions from step 1 being inputs. Elmachtoub and Grigas earlier proposed SPO where we take into account step 2 during step 1 by setting up an appropriate surrogate loss. Prior work has only established asymptotic guarantees for this learning problem and this paper presents first finite sample guarantees. The reviewers agree that this paper is well written and presents an important contribution that is of interest to a broader Neurips community.